# Structure and function of Semaphorin-5A glycosaminoglycan interactions

Gergely N. Nagy ◉[1,2,7] ✉, Xiao-Feng Zhao ◉[3], Richard Karlsson ◉[4], Karen Wang[3], Ramona Duman[5], Karl Harlos[1], Kamel El Omari ◉[5], Armin Wagner ◉[5], Henrik Clausen ◉[4], Rebecca L. Miller ◉[4] ✉, Roman J. Giger ◉[3,6] ✉ & E. Yvonne Jones ◉[1] ✉

Integration of extracellular signals by neurons is pivotal for brain development, plasticity, and repair. Axon guidance relies on receptor-ligand interactions crosstalking with extracellular matrix components. Semaphorin-5A (Sema5A) is a bifunctional guidance cue exerting attractive and inhibitory effects on neuronal growth through the interaction with heparan sulfate (HS) and chondroitin sulfate (CS) glycosaminoglycans (GAGs), respectively. Sema5A harbors seven thrombospondin type-1 repeats (TSR1-7) important for GAG binding, however the underlying molecular basis and functions in vivo remain enigmatic. Here we dissect the structural basis for Sema5A:GAG specificity and demonstrate the functional significance of this interaction in vivo. Using x-ray crystallography, we reveal a dimeric fold variation for TSR4 that accommodates GAG interactions. TSR4 co-crystal structures identify binding residues validated by site-directed mutagenesis. In vitro and cell-based assays uncover specific GAG epitopes necessary for TSR association. We demonstrate that HS-GAG binding is preferred over CS-GAG and mediates Sema5A oligomerization. In vivo, Sema5A:GAG interactions are necessary for Sema5A function and regulate Plexin-A2 dependent dentate progenitor cell migration. Our study rationalizes Sema5A associated developmental and neurological disorders and provides mechanistic insights into how multifaceted guidance functions of a single transmembrane cue are regulated by proteoglycans.

The Semaphorins (Sema) are a large family of secreted and membrane bound glycoproteins, originally identified as axon guidance molecules[1,2]. Subsequent studies revealed that semaphorins function more broadly, and are powerful regulators of cytoskeletal dynamics and adhesion, both in neuronal and non-neuronal cell types, important for tissue development and organism health[3–8]. Plexins (Plxn) are the major signal transducing receptors for Sema[9] and typically Sema dimers bring together two Plxn to trigger receptor signaling[10–13]. Members of the class 5 subfamily of semaphorins (Sema5) include vertebrate Sema5A and Sema5B and invertebrate Sema5C, closely related transmembrane proteins which boast the canonical extracellular sema and plexin-semaphorin-integrin (PSI) domains followed by seven thrombospondin type-1 repeats (TSR1-TSR7) and a short cytoplasmic domain[14,15]. Genome wide association studies revealed a

[1]Division of Structural Biology, Wellcome Centre for Human Genetics, University of Oxford, Oxford, UK. [2]Department of Applied Biotechnology and Food Science, Faculty of Chemical Technology and Biotechnology, Budapest University of Technology and Economics, Budapest, Hungary. [3]Department of Cell and Developmental Biology, University of Michigan Medical School, Ann Arbor, MI, USA. [4]Copenhagen Center for Glycomics, Department of Cellular and Molecular Medicine, Faculty of Health Sciences, University of Copenhagen, Copenhagen-N, Denmark. [5]Diamond Light Source, Harwell Science and Innovation Campus, Didcot, UK. [6]Department of Neurology, Ann Arbor, MI, USA. [7]Present address: Institute of Molecular Life Sciences, HUN-REN Research Centre for Natural Sciences, Budapest, Hungary. ✉e-mail: nagy.gergely.nandor@vbk.bme.hu; rmiller@sund.ku.dk; rgiger@med.umich.edu; yvonne.jones@strubi.ox.ac.uk

link between *SemaSa* mutations and autism spectrum disorder (ASD)[16–19].

Sema5s regulate neural circuit development in the mammalian retina, hippocampus, cortex, corticospinal tract, and zebrafish motor system[20–24]. Sema5A is a bifunctional guidance cue, and depending on context, exerts permissive or inhibitory properties toward developing axons and migrating cells[20,25–28]. Studies with Sema5A domain deletion mutants showed that the sema domain inhibits axon growth, while TSR1-7 are sufficient to promote axonal growth[20,27]. TSRs are a unique feature of Sema5s and enable interactions with heparan sulfate (HS) and chondroitin sulfate (CS) glycosaminoglycan (GAG) chains linked to the protein core of proteoglycans[27]. In cultured midbrain neurons, HS-GAGs are required cell autonomously for Sema5A to exert its growth promoting effects, whereas CS-GAGs in the local microenvironment are necessary to convert Sema5A from an attractive to an inhibitor guidance cue[27]. The structural basis of the TSR–GAG interactions and how a switch in GAG interactions alter Sema5A functions remains unresolved.

Here, we report the structural basis of the Sema5A-TSRs interaction with GAGs. X-ray crystallography identifies a novel fold for TSR4, critical for Sema5A dimerization and association with HS- and CS-GAGs. GAG modifications that support TSR4 association are uncovered in a cell-based binding assay and reveal preferential binding of specific HS- over CS-GAGs. Amino acid residues in the TSR4 fold necessary for GAG binding are independently validated by site-directed mutagenesis. In vivo, Sema5A:GAG interactions are necessary for the proper distribution of dentate progenitor cells in the mouse hippocampus, and we provide evidence that the Sema5A:GAG interaction regulates signaling strength through Plexin-A2. In summary, the study rationalizes Sema5A-associated developmental disorders and provides mechanistic insights into how the multifaceted functions of a single guidance cue are regulated by proteoglycans.

## Results

### Structural determinants of Sema5A-GAG interactions

Previous studies demonstrated that TSR1-4 of Sema5A (Sema5A$_{TSR1-4}$) binds GAGs[27]. By use of heparin affinity chromatography, we found binding of full-length ectodomain human Sema5A$_{sema-TSR1-7}$ and Sema5A$_{TSR3-4}$ constructs but not of Sema5A$_{sema-TSR2}$, indicating a decisive role of TSR3-4 for Sema5A-GAG interactions (Fig. 1a–c, Supplementary Fig. 1a–c). We determined four Sema5A$_{TSR3-4}$ crystal structures to high resolution (1.56–2.72 Å); apo state and complexes with nitrate, sulfate, and the heparin disaccharide mimetic sucrose octasulfate (SOS) (Figs. 1d–f, 2, Supplementary Fig. 2, Supplementary Table 1). Sema5A$_{TSR3-4}$ is a rod-like dimer (Fig. 1d); superposition of the various structures indicates TSR3-TSR4 inter-domain angle differences (Supplementary Fig. 3), which may contribute conformational flexibility to the Sema5A ectodomain. Beyond this variability, and the ligands with their immediate surroundings, these four structures are essentially identical, thus the following structural analyses focus on the highest resolution nitrate co-complexed structure unless indicated otherwise. The Sema5A TSR3 exhibits a typical TSR fold composed of a three-stranded antiparallel β-sheet with a bulged Strand A conformation, similarly as in Thrombospondin-1 (Fig. 1e, Supplementary Fig. 4 and ref. [29]). It is stabilized by stacked layers of Trp residues ($^{656}$WTGWGPW$^{662}$ from Strand A) and Arg and Gln residues ($^{674}$QARRR$^{678}$ from Strand B) capped by disulfide bonds (C665-C696), (C669-C701) and (C680-C686) at both ends, interconnecting strands A-C, B-C, and B-C, respectively. TSR domain tryptophans may be C-mannosylated[30], and we found well-resolved mannosyl residues linked to W656 and W659 (Fig. 1e, Supplementary Fig. 2f). Dimerization of the TSR3 domains is primarily supported by an intermolecular C689-C689′ disulfide as the dimer interface is limited with a few polar and van der Waals interactions contributing to a total buried surface area of 607 Å² (Supplementary Fig. 5).

Unexpectedly, the TSR4 dimer shows 3D domain swapping leading to a unique architecture with no structural homologs currently available in the PDB (Supplementary Fig. 4). Strand F crosses over to the other TSR4 protomer to form antiparallel strand interactions there with Strand E, leading to a dimer of antiparallel D′E′F and DEF′ β-sheets (Fig. 1f). The resultant intertwined dimer form is stabilized by an intermolecular C733-C733′ disulfide together with an exceptionally extensive network of salt bridges, hydrogen bonds, and van der Waals interactions contributing to a total buried surface area of 4475 Å² (Supplementary Fig. 5). The absence of inter-strand disulfide bonds in TSR4, a conserved feature of all other TSR domains within Sema5A (Supplementary Fig. 6a), enables this fold variation. TSR4 additionally harbors a C755-C763 disulfide, not shared with other TSR domains in Sema5A, that stabilizes a flexible loop at the C-terminal end of strand F. The TSR4 domain also features a Trp-Arg ladder albeit shorter than the one in TSR3. Tryptophans from Strand 1 ($^{710}$WTPW$^{713}$) are found interleaved between arginines from Strand 2 ($^{728}$RFR$^{730}$) and, intriguingly, R753′ from the domain-swapped Strand F′.

The novel architecture of TSR4 supports two symmetrically positioned ~20 × 20 Å-sized shallow cavities on the opposing sides of the Sema5A dimer. Positively charged and polar residues (K505, T732, K734, R747, R749) from both monomers contribute to each of these sites, together providing a notable positive electrostatic surface potential (Fig. 2a). Indeed, these charged cavities attracted various anions from the crystallization solutions with each cavity coordinating two nitrates and a single sulfate in the Sema5A$_{TSR3-4}$-NO₃ and Sema5A$_{TSR3-4}$-SO₄ structures, respectively (Fig. 2b, c, Supplementary Fig. 2b, c, Supplementary Fig. 7a, b, e, f). Co-crystallization with SOS also resulted in this heparin surrogate occupying one of these sites in the Sema5A$_{TSR3-4}$ dimer forming multiple electrostatic and polar interactions with the protein (Fig. 2d, Supplementary Figs. 2d, 7c, d, g–i). Collectively, the three co-crystal structures indicate that these cavities constitute the focus for Sema5A GAG interactions. There is an absence of clear electron density for key Sema5A GAG site sidechains, R747′ and R749′ in the apo Sema5A$_{TSR3-4}$ structure (Fig. 2e). The conformational flexibility of these residues is consistent with an adaptable binding site that can accommodate GAG binding partners in vivo. We further explored the GAG binding properties of the TSR4 domains by performing automated in silico rigid body docking of an IdoA(2S)-GlcNS(6S)-IdoA(2S)-GlcNS(6S) tetramer heparin fragment to the Sema5A$_{TSR3-4}$-NO₃ structure[31]. The representative positions for the two largest docked pose clusters are found within the positively charged cavities of the TSR4 dimer and are stabilized by multiple charged and polar protein-ligand interactions (Fig. 2f, g, Supplementary Fig. 8). A further ten docked dp4 positions, with less favorable binding energy scores, are anchored in one GAG site and extend towards the other. They together form a belt around the TSR4 dimer with c.a. 16 saccharide units mapping the perimeter of this putative extended GAG binding site.

### Sema5A boasts a unique GAG specificity

Next, we designed Sema5A charge-reversal mutants to interfere with GAG binding. We created Sema5A$_{TSR3-4}$ tandem double R747E/R749E and triple K734E/R747E/R749E mutants, and confirmed that both of them abrogated heparin binding (Supplementary Fig. 1c, d). Moreover, a Sema5A$_{sema-TSR1-7}$ ectodomain construct harboring the R747E/R749E tandem mutation also lost its ability to interact with heparin, confirming that TSR4 provides the only effective GAG binding site on Sema5A (Supplementary Fig. 1e). To assess whether mutating K734E/R747E/R749E abrogates interaction with endogenous GAGs present in neural tissue, we generated alkaline phosphatase tagged fusion proteins comprised of wildtype or mutated TSR(1-4). Recombinant protein was used for binding to neonatal brain tissue sections. Robust binding was observed for wildtype TSR(1-4), including inner retina, neocortex, and hippocampus. In marked contrast, mutated TSR(1-4)

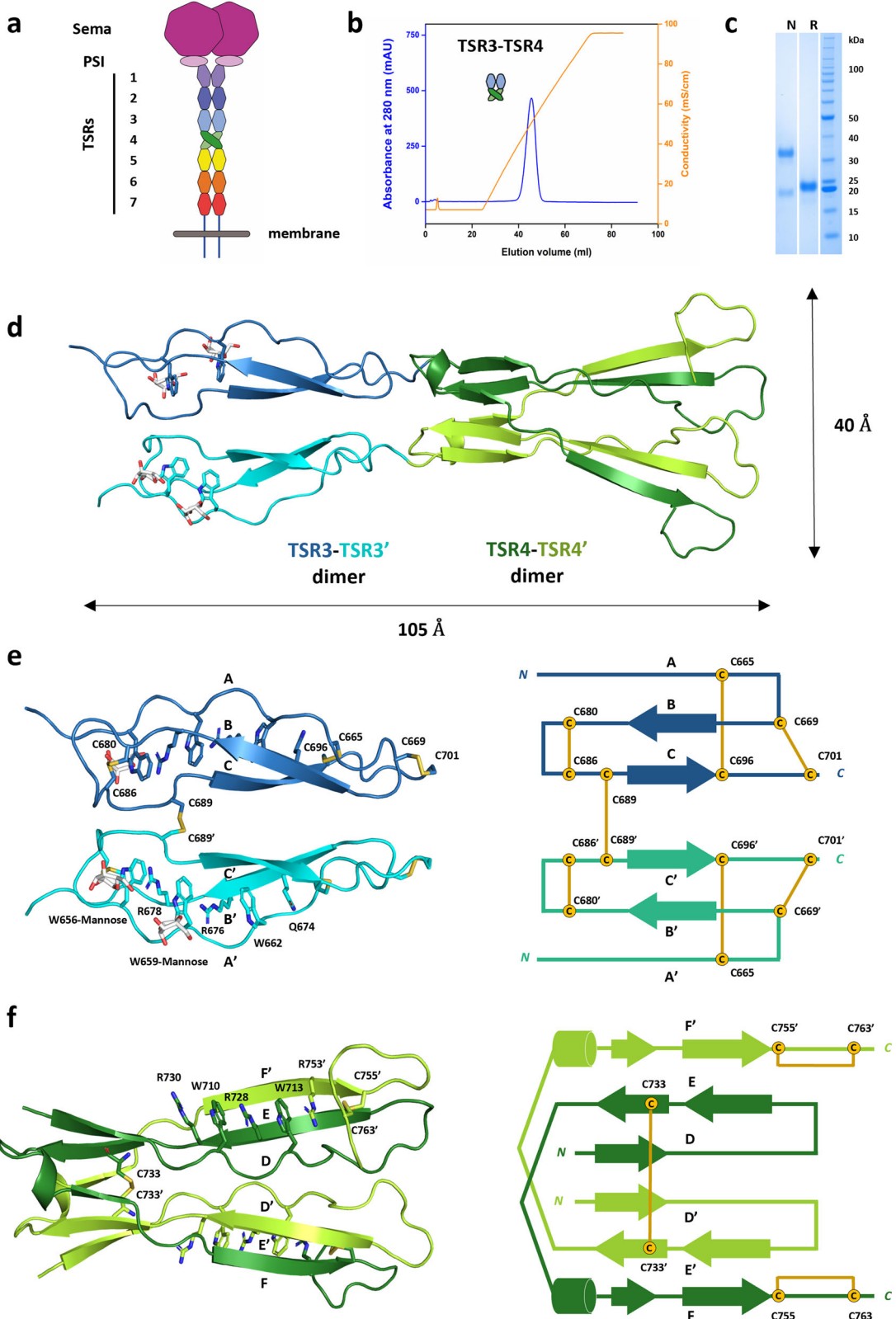

**Fig. 1 | Sema5A TSR3-4 is a disulfide-linked dimer with a unique 3D domain-swapped fold. a** Sema5A architecture. Sema, sema domain; PSI, plexin-semaphorin-integrin domain; TSR, thrombospondin-1 type repeat/domain. **b** Sema5A$_{TSR3-4}$ binds to a heparin column and elutes at ~500 mM NaCl, 10 mM HEPES pH 7.5 (46 mS/cm). mAU: milli absorbance units. **c** non-reducing (N) and reducing (R) SDS-PAGE analysis indicates an unexpected disulfide-linked dimer form. kDa: kilodaltons. Representative data is shown from 3 biological replicates. **d** Ribbon representation of the Sema5A$_{TSR3-4}$-NO$_3$ structure. Color code: Chain A

TSR3, blue; TSR4, dark green; Chain B TSR3, cyan; TSR4, light green. Inter-protomer disulfides are shown as sticks, and C-mannosyl modifications are colored by the element, carbon: white. **e, f** Cartoon and topology depiction of the TSR3 and TSR4 domain folds, respectively. *N* and *C* terminal ends of polypeptide chains are indicated by letters in italics. Inter- and intra-domain disulfides are shown as sticks (left) and lines (right). Trp-Arg ladders from both domains are shown as sticks. Source data are provided as a Source Data file.

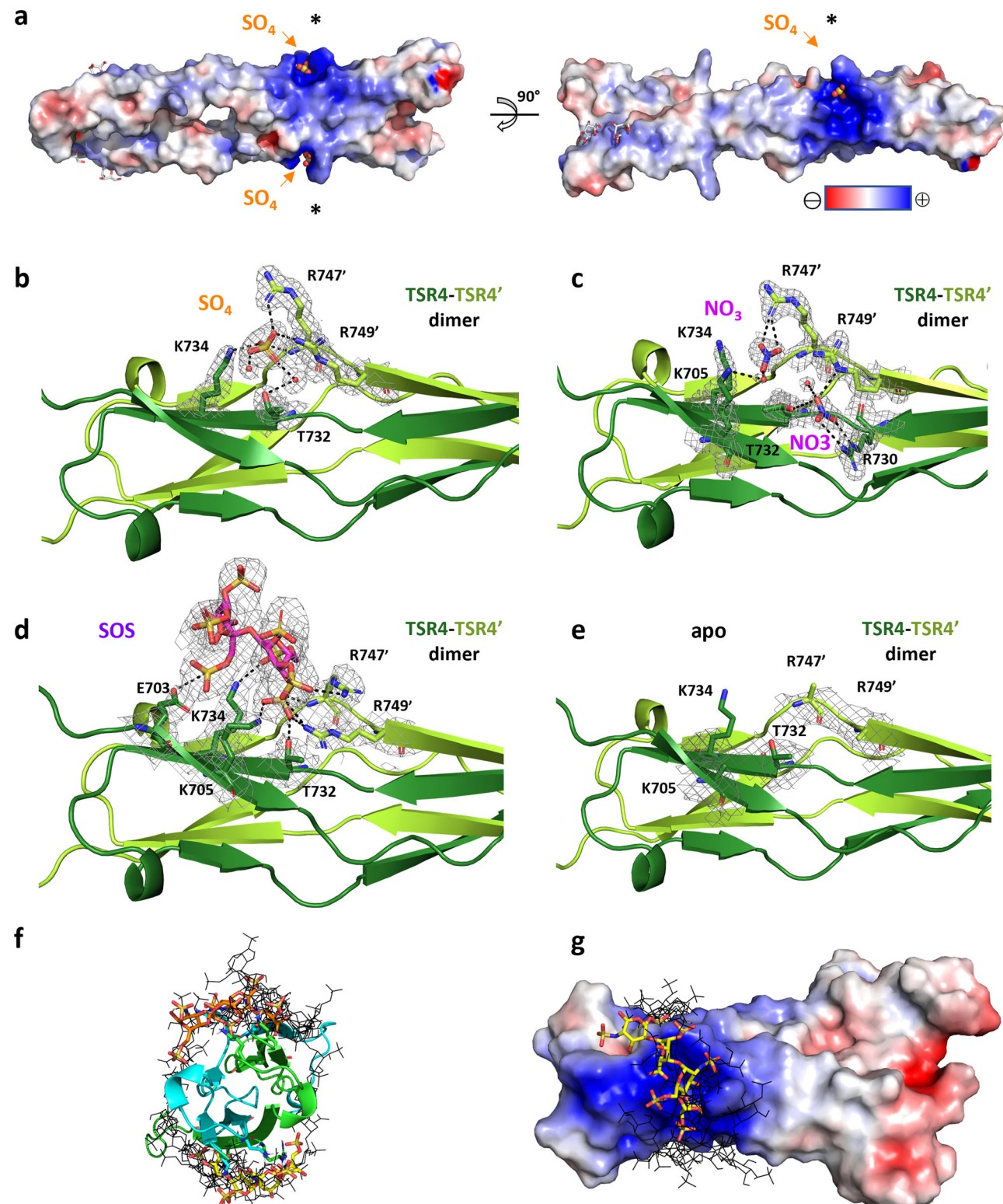

**Fig. 2 | Structural characterization of the Sema5A GAG site. a** Electrostatic surface potential for the Sema5A$_{TSR3-4}$-SO$_4$, calculated by the Adaptive Poisson-Boltzmann Solver (ABPS) and visualized on a red (−4 kbT/ec) to blue (+4 kbT/ec) color range. Each of the two positively charged cavities (marked with asterisks) formed by TSR4 dimer accommodates a sulfate ion (sticks). Snapshot of the GAG site within TSR4; (**b**), Sema5A$_{TSR3-4}$-SO$_4$, (**c**), Sema5A$_{TSR3-4}$-NO$_3$, (**d**), Sema5A$_{TSR3-4}$-SOS, (**e**), Sema5A$_{TSR3-4}$-apo. Residues are shown as sticks and the refined 2F$_o$-F$_c$ electron density map is shown at 1.0 root mean square deviation contour level. H-bonds are indicated as dashed lines. **f** Cross-section view of TSR4 domain from

Sema5A$_{TSR3-4}$-NO$_3$ rigid body docked with a heparin dp4 IdoA(2S)-GlcNS(6S)-IdoA(2S)-GlcNS(6S) tetrasaccharide using ClusPro. Docked ligand poses representing the largest and second-largest clusters are shown as sticks, color code: yellow and orange carbons, respectively. Further 11 docked dp4 poses are shown as black lines. **g** Docked dp4 pose representing the largest cluster is accommodated in the GAG binding site of Sema5A. The TSR4 dimer is represented by its electrostatic surface potential calculated by ABPS and visualized on a red (−4 kbT/ec) to blue (+4 kbT/ec) color range. Source data are provided as a Source Data file.

failed to support binding (Supplementary Fig. 12f, g). This demonstrates that K734, R747, and R749 are necessary for binding to GAGs present in vivo.

Next, we measured the binding affinity of Sema5A$_{sema-TSR1-7}$ to surface immobilized heparin, chondroitin sulfate A (CS-A), and chondroitin sulfate E (CS-E) using BioLayer Interferometry (BLI). The Sema5A$_{sema-TSR1-7}$ – heparin interaction exhibited the highest affinity ($K_d = 0.32 \mu M$), while Sema5A$_{sema-TSR1-7}$ – CS-E interaction had a ~10-fold lower affinity ($K_d = 2.7 \mu M$). Introduction of the R747E/R749E mutations into Sema5A$_{sema-TSR1-7}$ largely abolished CS-E binding and we additionally failed to observe binding between Sema5A$_{sema-TSR1-7}$ and CS-A (Fig. 3a, Supplementary Fig. 1f–i). These data indicate that Sema5A GAG interactions are specific and selective for HS/CS and sulfation patterns of GAGs.

HS and CS chains are composed of repeating disaccharide units containing hexosamine and hexuronic acid building blocks which are site-specifically modified by deacetylation, sulfation and epimerization, providing heterogeneous and diversified GAG forms with an overall high negative charge[32]. To dissect Sema5A GAG binding specificities, we utilized a genetically engineered library of isogenic Chinese Hamster Ovary (CHO) cells, each with different capacities for GAG biosynthesis due to gene knock-out (KO) and knock-in (KI) of GAG biosynthesis genes (Fig. 3, Supplementary Fig. 1k, l). This CHO library serves as a cell-based display of GAG variants, also referred to as the GAGOme[33,34], in which the effects of loss/gain of genes on binding of proteins provide indications of the GAG genes and biosynthetic pathways required for the observed binding, and this information may be interpreted to indicate the GAG features involved in binding. CHO WT cells produce rather heterogenous HS and CS GAGs with a repertoire of modifications, and we have previously demonstrated that loss/gain of binding as a result of KO/KI of GAG genes can be used to dissect binding specificities of GAG binding proteins[33,34]. Flow cytometry binding assays with secreted Sema5A deletion mutants revealed that Sema5A$_{TSR3-4}$ bound to WT CHO cells, while the Sema5A$_{TSR3-4}$ R747E/R749E mutant did not bind (Fig. 3c). Sema5A$_{TSR3-4}$ binding to CHO cells was abolished by KO of *B4galt7*, which completely abrogates both HS and CS biosynthesis (HS + CS KO) by disruption synthesis of the common tetrasaccharide linker on which GAG chains are build (Fig. 3c). The Sema5A$_{TSR3-4}$ binding was shown to require HS and not CS as KO of *Extl3* (HS KO) completely abolished binding, while KO of *CSGalNAcT1/CSGalNAcT2/Chsy1* (CS KO) did not. Since CHO WT cells mainly produce the CS-A variant of CS[33], which in contrast to CS-E did not bind Sema5A$_{TSR3-4}$ in our BLI assay, we also tested Sema5A$_{TSR3-4}$ binding to CHO cells with KI of CHST3 to increase CS 6-O-sulfation[33] and KI of CHST15 (cells producing tiny quantities of CS-E[33]), however, the respective binding signals were no greater than that for CHO WT (Supplementary Fig. 1m). These data suggest that either Sema5A does not interact with the set of CSPGs displayed at the CHO cell surface, or, that there is a pronounced preference to bind HS over CS epitopes. To further dissect the HS modifications that support Sema5A$_{TSR3-4}$ binding we targeted *N*-deacetylation/*N*-sulfation by KO of *Ndst1/2* (prerequisite for further HS modifications[35]), GlcA/IdoA epimerization by KO of *Glce*, 2-O-sulfation by KO of *Hs2st1*, and 6-O-sulfation by KO of *Hs6st1/2/3*, which all abrogated binding, thus clearly indicating that the observed binding was to HS (Fig. 3b, c). We also tested CHO cells with enhanced *N*- and 6-O-sulfation by KI of NDST2 or HS6ST1, respectively, as well as cells with introduced 3-O-sulfation (absent in CHO WT cells) by KI of HS3ST1 and HS3ST5 (Supplementary Fig. 1m). Since none of these introduced features affected binding, we conclude that Sema5A binding relies on HS with a complex sulfation pattern of *N*-, 2-O-, and 6-O-sulfate groups as well as IdoA residues.

## HS-GAGs oligomerize Sema5A

HS can act as a concatenator that oligomerizes surface receptors or ligands providing an additional layer of regulation[36]. To assess whether

such effects modulate Sema5A function, we analysed its interactions with heparin and HS using mass photometry (Fig. 4a–c, e–h and ref. 37). This demonstrated Sema5A$_{sema-TSR1-7}$ as a single peak with a molecular mass of ~250 kDa corresponding to a dimer (Fig. 4b). The titration of Sema5A$_{sema-TSR1-7}$ with heparin (stoichiometry ranging from 1:50 to 1:0.25) resulted in the appearance of masses correlating with multiples containing up to 4 Sema5A dimers (Fig. 4c, Supplementary Fig. 9a–g). We found that the stoichiometry of Sema5A$_{sema-TSR1-7}$ to heparin played a key role in oligomerization, as excess heparin (1:50) prevented oligomerization and low quantities of heparin (1:0.25) demonstrated minimal induction of Sema5A$_{sema-TSR1-7}$:heparin oligomerization. The observed Sema5A multimerization was not found with CS-E (Sema5A$_{sema-TSR1-7}$: CS-E stoichiometry range of ~1:50 to 1:0.25) (Fig. 4d, Supplementary Fig. 10a–g). To further study the HS features involved in Sema5A multimerization, we used HS isolated from cell lysates of engineered CHO cells (Fig. 4e–h). HS derived from WT CHO cells clearly supported Sema5A multimerization with the appearance of a mass corresponding to HS with two dimer Sema5A molecules (Fig. 4e). Interestingly, HS isolated from CHO cells with introduced 3-O-sulfation (HS3ST1 KI or HS3ST5 KI in CHO cells without contaminating CS (Csgalnact1/2/Chsy1 KO)) appeared to support further multimerization with appearance of a mass corresponding to HS with three Sema5A molecules (Fig. 4f, g). Additionally, HS isolated from CHO cells with KI of NDST2 appeared to reduce multimerization (Fig. 4h). We did not analyse HS from the engineered CHO cells with KO of GAG genes as these did not support binding (Fig. 3). Our results with HS isolated from engineered CHO cells clearly confirm that Sema5A binds HS and can oligomerize on HS chains, further studies are needed to dissect the minimum epitopes.

We further assessed the HS length requirements for Sema5A multimerization by addition of size-fractionated heparin oligomers ranging from tetrasaccharide (degrees of polymerization (dp)4) to icosasaccharide (dp20). These oligosaccharides did not produce multimeric Sema5A showing that a minimal length of GAG chains is required (Supplementary Fig. 9h–l). This is consistent with a > 70 Å steric requirement for multimerization imposed by the Sema5A sema domain diameter; the length of a fully extended dp18 heparin oligosaccharide is ~72 Å (cf. PDB: 3IRI) (Fig. 4i). Sema5A GAG multimerization analysis in Multi-Angle Light Scattering (MALS) provided consistent results with some multimerization effect observed for the Sema5A$_{sema-TSR1-7}$ + dp30 heparin sample (Fig. 4j). We additionally tested whether CS-E could compete with Sema5A heparin interactions and found that neither CS-E and Sema5A$_{sema-TSR1-7}$ preincubation before heparin addition, nor CS-E addition to a heparin-Sema5A$_{sema-TSR1-7}$ mix could suppress heparin-dependent Sema5A multimerization (Supplementary Fig. 10h–j). Together, these results indicate that HS is the primary GAG binding partner for Sema5A, and unlike CS, mediates oligomerization.

Next, we used isothermal titration calorimetry (ITC) to further characterize the interaction of Sema5A$_{sema-TSR1-7}$ with size-fractionated heparin oligosaccharides, ranging from dp4 to dp20, fragments too short to support Sema5A multimerization (Fig. 4k, Supplementary Fig. 11). The data reveal a trend of increasing binding enthalpy with heparin size, binding affinity reached a maximum value of 3–4 μM for dp12-dp20. Titration data for the dp12 and longer heparin oligosaccharides was compatible with a model of a single heparin binding to Sema5A dimer, but not with a 2:1 heparin vs Sema5A dimer stoichiometry (Supplementary Fig. 11i–k). These data, together with the rigid body docking results (Fig. 2), suggest that 12-mer and longer heparin oligosaccharides may be able to access both GAG sites of a Sema5A dimer resulting in an increased binding affinity. Our results support a model where a ~12–16-mer HS epitope is required for binding to a Sema5A dimer, accessing both of its GAG binding sites, and a further minimum 18-mer linker is required to reach another Sema5A molecule, enabling increased functional affinity through multimerization (Fig. 4i).

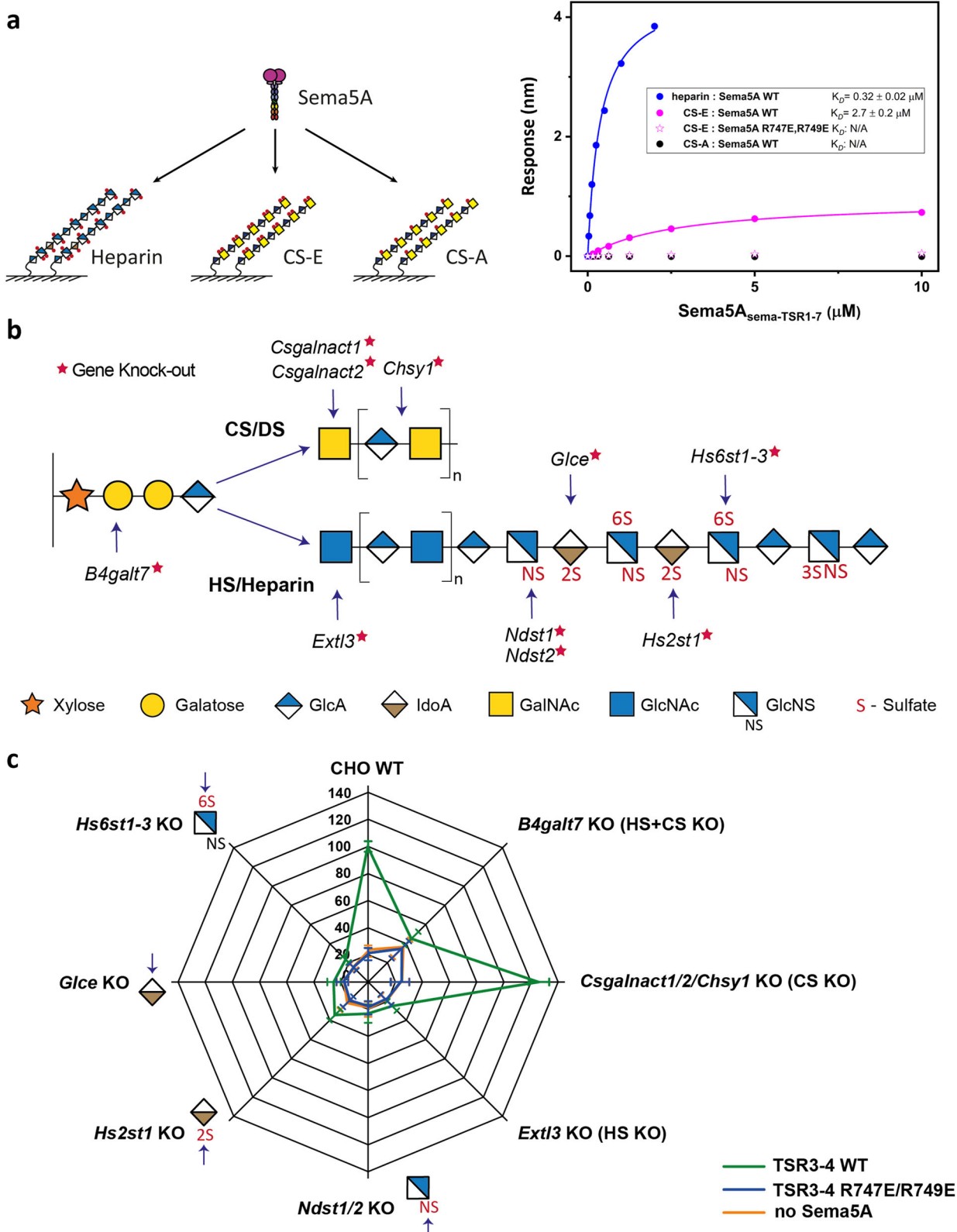

**Fig. 3 | Dissecting the specificity of Sema5A interactions with GAGs. a** BLI binding analysis to characterize the Sema5A interaction with heparin, CS-E, and CS-A. A schematic of the experiment and the calculated apparent dissociation constant ($K_D$) values are shown. **b** A subset of the genes controlling HS or CS chain elongation and modifications that were engineered with KO (red star) in CHO cells and used in this study. **c** Sema5A$_{TSR3-4}$ binding to genetically engineered CHO cell lines generated by KO of genes encoding GAG biosynthesis enzymes. The radar chart shows the relative mean fluorescence intensity (MFI) from flow cytometry (WT cells: 100) after genetic KO of the indicated genes. HS and CS disaccharide composition for each cell line is described in refs. 33,34. Source data are provided as a Source Data file.

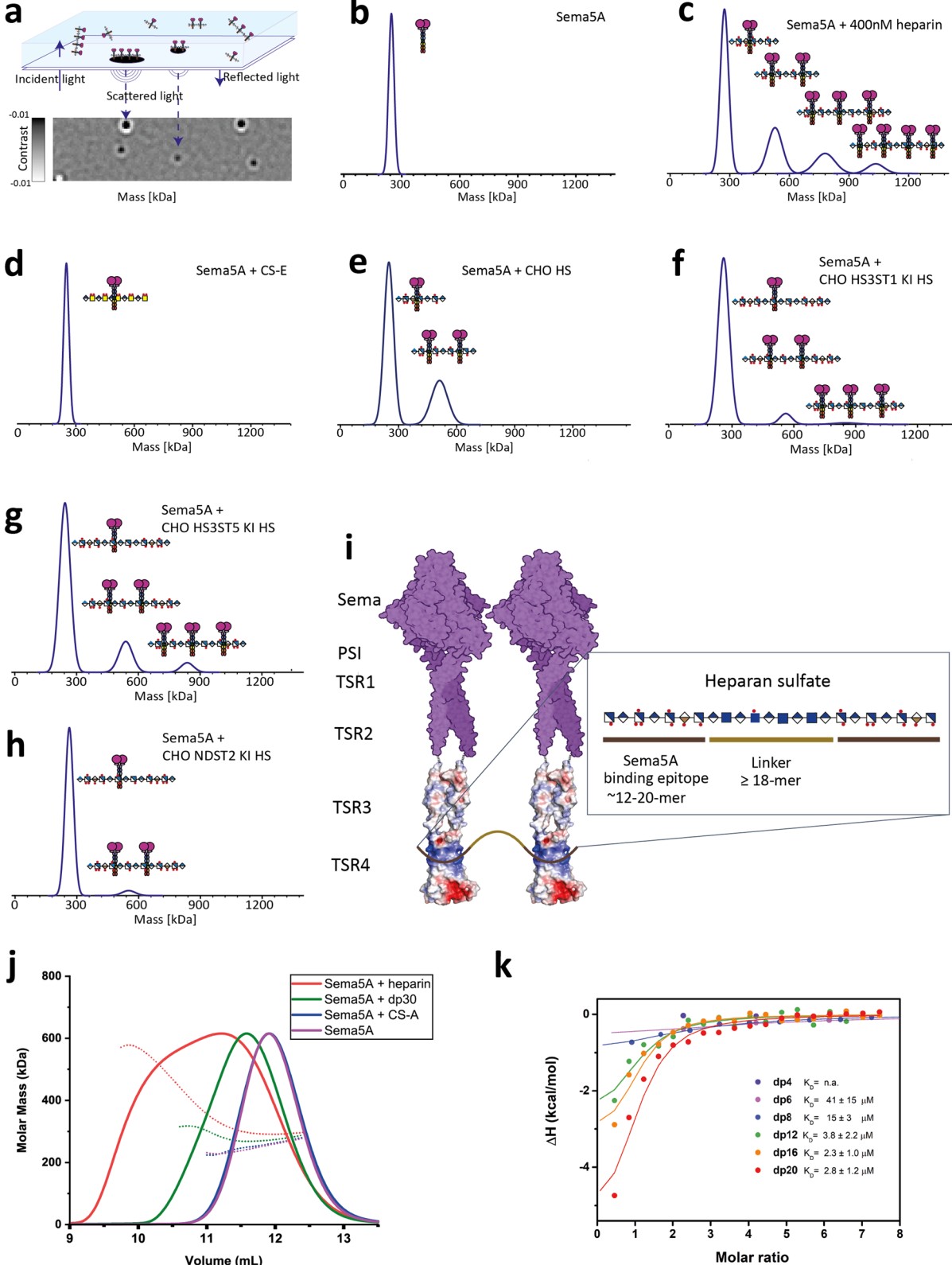

## Sema5A-GAG interaction does not regulate dendritic spines

To assess the functional significance of the Sema5A-GAG interaction in vivo, we used CRISPR/Cas9 based gene editing to generate *Sema5a(R747E;R749E)* mice, deficient for GAG binding, hereafter referred to as *Sema5a^GAG* mice (Supplementary Fig. 12). Two independent *Sema5a^GAG* lines were established and analysed. Similar to *Sema5a^-/-* mice, homozygous *Sema5a^GAG/GAG* mice are viable into

adulthood, fertile, and at the gross anatomic level, show no obvious defects (Supplementary Fig. 12c–e). Sema5A is abundantly expressed by developing and adult dentate granule cells (GC). We previously found that in *Sema5a^-/-* mice, there is a significant increase in dendritic spines and excitatory synapse density in GC[22]. To assess GC spine density in *Sema5a^GAG/GAG* mutants, mice were crossed onto a *Thy1-EGFPm* background (*Sema5a^GAG/GAG;Thy1-EGFP*) for selective labeling of

**Fig. 4 | Analysis of Sema5A: GAG interactions and oligomerization. a** Graphic schematics of principles of mass photometry analysis. Mass photometry with mass distributions of Sema5A and GAG chains. (**b**) Sema5A$_{sema-TSR1-7}$ alone, (**c**) Sema5A$_{sema-TSR1-7}$ mixed with heparin (1:4 stoichiometric ratio), (**d**) Sema5A$_{sema-TSR1-7}$ mixed with CS-E (1:4) (**e**) Sema5A$_{sema-TSR1-7}$ mixed with HS (1:4) derived from CHO cells without CS (CSGalNAcT1/2/Chsy1 KO (CS KO)), (**f**) Sema5A$_{sema-TSR1-7}$ mixed with HS (1:4) derived from CHO cells with HS3ST1 KI and without CS, (**g**) Sema5A$_{sema-TSR1-7}$ mixed with HS (1:4) derived from CHO cells with HS3ST5 KI and without CS; (**h**) Sema5A$_{sema-TSR1-7}$ mixed with HS (1:4) derived from CHO cells with NDST2 KI and without CS, Contrast measurements were converted to mass using calibration standards of known proteins. **i** Illustration of Sema5A multimerization by heparan sulfate. A chimera Sema5A$_{sema-TSR1-4}$ model is shown from the combination of a dimer human Sema5A$_{sema-TSR1-2}$ model created by Alphafold2[69] (magenta surface) and Sema5A$_{TSR3-4}$-NO$_3$ crystal structure (visualized as Fig. 2a). The arrangement of the Sema5A dimers in a parallel fashion designates the minimum distance requirement necessary for multimerization. Illustration is created with BioRender.com. **j** MALS analysis of Sema5A$_{sema-TSR1-7}$ multimerization with GAG preparations. **k** ITC binding isotherms (normalized heats versus molar ratio) for the interactions of size-defined heparins ranging dp4-dp20 to Sema5A$_{sema-TSR1-7}$. Source data are provided as a Source Data file.

a subset of GC[22]. The overall morphology of the hippocampus, including GC dendrites in the dentate molecular layer and GC projections, called mossy fibres, to the hilus and CA3 region, appear normal and are comparable to parallel processed *Sema5a$^{+/+}$;Thy1-EGFP* control mice (Fig. 5a, b). Next, we used confocal microscopy to analyse dendritic spine density within the rostral hippocampus located in the middle and outer thirds of GC dendrites (Fig. 5a, b). In postnatal day (P) 30 *Sema5a$^{+/+}$* mice, dendritic spine density (1.8 ± 0.046 spines/μm) was not significantly different from *Sema5a$^{GAG/GAG}$* mice (1.9 ± 0.103 spines/μm) (Fig. 5c). This suggests that the Sema5A-GAG interaction does not influence GC dendritic spine density.

### Sema5A-GAG interaction controls progenitor cell migration

*Sema5a* and its receptor, *Plxna2*, have previously been shown to regulate progenitor cell migration along the dentate migratory stream and distribution in the subgranular zone (SGZ) of the developing dentate gyrus[38]. Gene expression analysis revealed that both *Sema5a* and *Plxna2* are strongly expressed by immature GC (Supplementary Fig. 13). In addition, several HSPGs (*Gpc5, Gpc6, Sdc2, Sdc4*) and CSPGs (*Bcan, Ncan*) are expressed by GFAP$^+$ cells that form the radial glial scaffold along which immature GC migrate (Supplementary Fig. 13). To assess progenitor cell distribution and proliferation in the P14 dentate gyrus of *Sema5a$^{GAG/GAG}$* mice, we used BrdU pulse-labeling. In coronal sections through the rostral hippocampus, many BrdU$^+$ cells are found in the dentate hilus of wildtype (*Sema5a$^{+/+}$*) mice. The majority of BrdU$^+$ cells are confined to the SGZ, the neurogenic niche lining the inner rim of the granule cell layer, with few BrdU$^+$ cells outside of the SGZ within the deep hilar region (Fig. 5d). In *Sema5a$^{GAG/GAG}$* littermates, the number of BrdU$^+$ cells in the hilus is significantly reduced compared to parallel processed *Sema5a$^{+/+}$* mice (Fig. 5e, i). These studies show that the Sema5A-GAG interaction controls the number of progenitor cells in the SGZ in vivo.

### Disrupting Sema5A-GAG binding mimics Sema5A loss of function

Similar to *Sema5a$^{-/-}$* mice, *Plxna2$^{-/-}$* mice show defects in dentate progenitor cell migration and distribution within the hilus[38]. BrdU pulse-labeling of P14 *Plxna2$^{+/-}$* mice revealed haploinsufficiency for progenitor distribution in the SGZ (Fig. 5f). The total number of BrdU$^+$ cells in the dentate is comparable to wildtype mice, however there are fewer BrdU labeled cells in the SGZ and significantly more in the deep hilus (Fig. 5f, i, j). In transheterozygous mice, lacking one allele of *Plxna2* and one allele of *Sema5a* (*Plxna2$^{+/-}$;Sema5a$^{+/-}$*), defects observed in *Plxna2 +/-* mice are rescued (Fig. 5g). Because loss of the Sema5A-GAG association may result in Sema5A gain-of-function or loss-of-function, depending on types of HS and CS GAGs in the micro-environment, we wondered whether on a sensitized *Plxna2$^{+/-}$* background, reduction of the Sema5A-GAG association rescues or worsens defects observed in *Plxna2$^{+/-}$* mice. To address this question, we generated compound heterozygous mice (*Plxna2$^{+/-}$;Sema5a$^{GAG/+}$*) and assessed distribution of progenitor cells following BrdU pulse labeling (Fig. 5h). Quantification of BrdU$^+$ cells in *Plxna2$^{+/-}$;Sema5a$^{+/-}$* and *Plxna2$^{+/-}$;Sema5a$^{GAG/+}$* mice, revealed comparable numbers and distribution, indicating that mutation of the Sema5A GAG binding site results in a loss of function allele.

(Fig. 5i, j). Our studies also suggest that the TSR-GAG interaction of Sema5A influences PlxnA2 mediated progenitor cell distribution.

## Discussion

Our structural analysis of the Sema5A TSR3-4 region reveals a novel covalently linked dimer folding unit in TSR4 that presents two symmetrically located GAG binding sites, each site formed by the juxtaposition of basic residues from both monomers. Sequence comparisons show that both the TSR4 architecture and the charged residues at this GAG binding site are highly conserved across vertebrate Sema5 family members and species, suggesting a common GAG binding mode (Supplementary Fig. 6b). HSPGs and CSPGs regulate numerous cell surface signaling events with typically different effects on cell function. Sema5A, similarly to receptor protein tyrosine phosphatases[39,40], itself confers such a bifunctional signaling effect by interacting with HSPGs and CSPGs. Our study resolves several aspects of differential Sema5A interactions to these GAGs (Supplementary Fig. 14). We show that HS- and CS-GAGs can interact with the same GAG binding region on Sema5A and that disruption of the GAG binding site abrogates Sema5A TSR(1-4) binding to proteoglycans in vivo. The intrinsic chemical differences of specific GAGs, including GAG chain flexibility modulated by epimerization, and the number and distribution of highly sulfated binding epitopes along the chains, could form the basis of difference in the affinity and molecular mechanism of Sema5A interactions with these two GAG classes. Nevertheless, our cell-based and in vitro binding analyses unanimously indicate that HS is a much more potent binding partner for Sema5A. We show that sulfated HS epitopes with NS, 2S, 6S modifications are required for Sema5A binding, accessing both GAG sites of a Sema5A dimer, and that HS-GAGs can multimerize Sema5A. Such long HS chains may have the potential to bridge between two cells decorated with Sema5A, providing an adhesive effect, which could serve as a molecular mechanism for the observed Sema5A-HS dependent neurite axon fasciculation and growth permission[27]. A fasciculation mechanism that relies on Sema5A-HSPG interactions could further contribute to the developmental delay of motoneurons observed in zebrafish lacking the Sema5A TSR region[20]. It remains to be addressed whether extracellular HS 6-O-endosulfatases which contribute to the murine nervous system development[41] could dynamically regulate Sema5A-HS signaling in neuronal development, similarly to their effect on morphogens and signaling molecules, reviewed in[42]. According to our cell-based glycan array data, removal of 6-O-sulfate from HS by Sulfs would attenuate Sema5A interactions and likely disrupt Sema5A-HS multimerization.

GAGs presented on the cell surface and extracellular matrix may regulate the stoichiometry of receptor complexes for guidance cues during axon pathfinding, however few in vivo analyses exist to support this view. One example is motor axon defasciculation by Drosophila Sema1A through PlexA, a guidance event that critically depends on the presence of secreted perlecan (encoded by *Hspg2*)[43]. Accordingly, an important question is whether Sema5A-GAG interactions augment or attenuate Sema5A signaling through PlxnA receptors[22,38]. We find that depending on context, dendritic spine density versus progenitor cell migration, disruption of the Sema5A-GAG interaction mimics *Sema5a* loss-of-function. Progenitor cell migration defects observed in

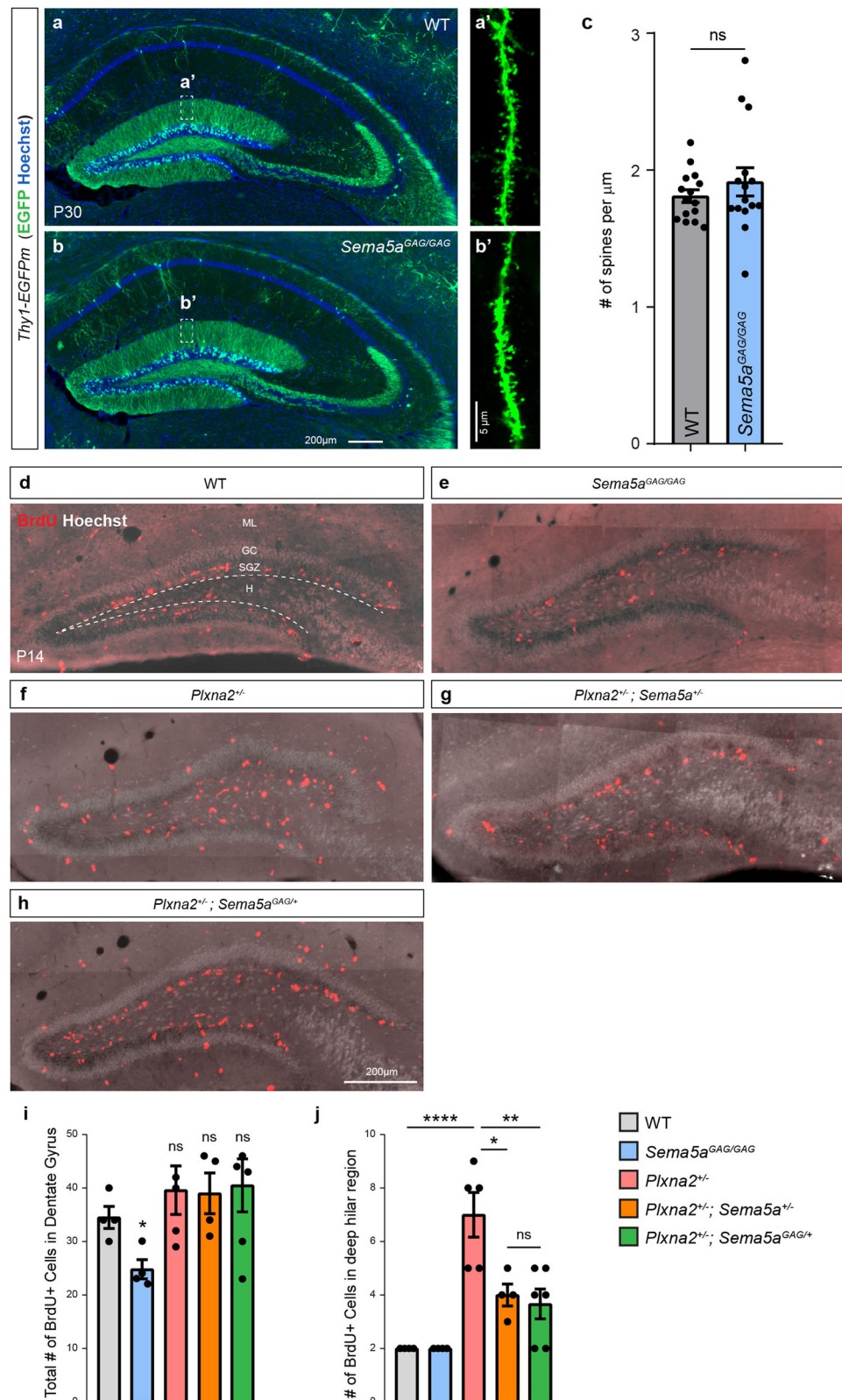

*Plxna2*[+/-] mice are sensitive to reducing the dose of *Sema5a*[GAG] and *Sema5a*. Loss of either allele rescues defects observed in *Plxna2*[+/-] mice. Migrating dentate progenitor cells express Sema5A and likely interact with HSPGs and CSPGs on radial glial cells, along which progenitor cells migrate. Based on studies with midbrain neurons, the Sema5A-HSPG interaction is permissive, and the Sema5A-CSPG interaction is non-permissive[27]. Directed progenitor migration requires the coordinated action of membrane extension, adhesion, translocation, and breaking of adhesion. Disruption of the Sema5A-CSPG interaction may result in too much adhesion and cause progenitor cells to get stuck, conversely loss of the Sema5A-HSPG interaction may result in too much repulsion preventing membrane extension and directional migration, both resulting in reduced cell migration. There is limited information available on how sema and TSR domain interactions of

**Fig. 5 | The Sema5A GAG interaction regulates stem cell distribution in the dentate in a *Plxna2* dependent manner. a, b** Coronal brain sections through the P30 hippocampus of *SemaSa⁺/⁺;Thy1-EGFPm* and *SemaSaᴳᴬᴳ/ᴳᴬᴳ;Thy1-EGFPm* mice, showing EGFP-labeled granule cells and their projections. **a′, b′** Higher magnification images of GC dendrites from the middle one-third of the dentate molecular layer, the corresponding areas are highlighted with dotted lines in (**a, b**). **c** Quantification of GC dendritic spine density (*n* = 3 mice per genotype, with 5 dendritic segments analysed per mouse). Coronal brain sections through the dorsal DG of P14 (**d**) WT (*n* = 4), (**e**) *SemaSaᴳᴬᴳ/ᴳᴬᴳ* (*n* = 4), (**f**) *Plxna2⁺/⁻* (*n* = 5), (**g**) *Plxna2⁺/⁻;SemaSa⁺/⁻* (*n* = 4) and (**h**) *Plxna2⁺/⁻;SemaSaᴳᴬᴳ/⁺* (*n* = 6) mice, stained with anti-BrdU. In (**d**), the border between the SGZ and the deep hilus (H) is marked with a dotted line. **i** Quantification of total number of BrdU⁺ cells in the DG per tissue section. Two-tailed unpaired Student's *t* test, *\*p* = 0.0119. **j** Quantification of BrdU⁺ cells in deep hilar region within the DG. Data are presented as mean ± SEM. One-way ANOVA multiple comparisons, *\*p* = 0.0107, *\*\*p* = 0.0017 and *\*\*\*\*p* < 0.0001. ns, not significantly. DG, dentate gyrus; GC granule cell, H Hilar region, ML molecular layer, SGZ subgranular zone. Scale bar: 200 μm in (**a, b, d–h**); 5 μm in (**a′, b′**). Source data are provided as a Source Data file.

Sema5A influence each other's signaling propensity. Possible mechanisms include contributions of GAG interaction dependent Sema5A clustering to Sema5A-PlxnA2 signaling modulation. Further studies are required to investigate whether Sema5A and its GAG interaction partners are arranged on the same or opposing cell surfaces, and how this arrangement modulates PlxnA2 signaling and progenitor cell migration.

Deleterious missense mutations in the SEMA5A gene are associated with various types of developmental and neurological disorders in humans. A microdeletion that results in loss of the last five TSRs (including TSR4) and cytoplasmic region in a boy with autism spectrum disorder (ASD) and speech delay[17]. A p.(Arg676Cys) mutation was identified in another patient diagnosed with ASD[17]. The Sema5A$_{TSR3-4}$ structure revealed that the conserved arginine residue R676 forms a central hub in the Trp-Arg ladder of TSR3, embedded in a W559-R676-W662 cation-π sandwich (Fig. 1f, Supplementary Fig. 2f). R676 additionally provides H-bond interactions to E692, and C-mannose coupled to W659, further contributing to structural stabilization. The R676C dissimilar mutation likely compromises folding, and stability of the protein based on a small-scale protein expression test of Sema5A$_{sema-TSR1-7}$ constructs (Supplementary Fig. 2g).

In summary, we report an in-depth Sema5A structure-to-function analysis that addresses the structural basis of how GAG/proteoglycan binding to Sema5A is accomplished, what type of GAG modifications are preferentially recognized by Sema5A, the functional significance of the Sema5A:GAG interaction in vivo. This study provides a hereto rare exemplar of how multifaceted guidance functions are regulated by proteoglycans.

## Methods

### Cell lines
HEK293T cells (ATCC, cat# CRL-3216) used in this study were cultured in Dulbecco's Modified Eagle Medium, DMEM, supplemented with 10% FBS, 1% NEAA and 1% L-Glutamine at 37 °C and 5% CO$_2$. Genetically engineered CHO cells, B4GalT7 KO, Csgalnat1/2/Chsy1 KO, Extl3 KO, Ndst1/2 KO, Hs2st1 KO, Glce KO, Hs6st1-3 KO, NDST2 KI, HS3ST1 KI Csgalnact1/2/Chsy1 KO, HS3ST5 KI Csgalnact1/2/Chsy1 KO, CHST3 KI, CHST15 KI were maintained in suspension culture using an equal mix of EX-CELL® CD CHO Serum-Free Medium (#14360 C, Sigma-Aldrich) and BalanCD CHO Growth A medium (#91128, Fujifilm), supplemented by 2% GlutaMAX™ (#35050061, Gibco). These CHO cell lines were generated and are maintained in the Copenhagen Centre of Glycomics, the cell plasmids and gRNA constructs designed for each CHO cell are described in refs. 33,34. Protein expression was induced by transfection with plasmid DNA and polyethyleneimine as the transfection reagent.

### Glycosaminoglycans from commercial sources
Unfractionated heparin Na salt, sourced from porcine intestinal mucosa was purchased from Merck, product code: 375095; Mw: ~6−30 kDa as stated by the vendor. CS-E from shark cartilage was purchased from Iduron (Manchester, UK), product code: GAG-CSE01. CS-A (sodium salt, from bovine trachea) was purchased from Merck,

product code: 27042. Size-defined heparin oligosaccharides were purchased from Iduron Ltd (Manchester, UK). The dp4-dp20 heparin oligosaccharides (where dp denotes the degree of polymerization, within, 'n' is the number of monosaccharides), product codes: HO04-20, were prepared by partial heparin lyase digestion of high quality heparin followed by high resolution gel filtration. Note that uronic acid (HexA) at the non-reducing end of the oligosaccharides has a C4-C5 double bond as a result of the endolytic action of bacterial heparin lyase. Average values for molecular weights, that includes the contribution of ammonium counterions, are provided by the vendor.

### Constructs and cloning
The constructs used in this study were cloned into pHLsec[44] or pcDNA 1.1 variant vectors with relevant tags for expression in cell lines. A construct of human Semaphorin-5A (Sema5A) (UniProtKB: Q13591), Sema5A$_{sema-TSR1-7}$, (residues 23E-944S) was cloned into the pHLsec vector in-frame with a C-terminal hexahistidine (His$_6$) tag. Constructs of Sema5A$_{sema-TSR2}$ (residues 23E-651P) and Sema5A$_{TSR3-4}$ (r. 652P-765T) were cloned into the pHLsec vector in-frame with a C-terminal 3C-Avi-His$_6$ tag. For protein purification, we used pHLsec vectors which also code for a C-terminal His$_6$-tag or a C-terminal 3C-Avi-His$_6$-tag, for BLI and GAGOme assay we used a C-terminal 3C-Avi-His$_6$-tag. A construct of human Sema5A Sema5A$_{sema-TSR1-7}$, (residues 23E-944S), harboring the R676C mutation was gene synthesized by Genescript and was subcloned into pHLsec vector. Primers used for cloning and mutagenesis are summarized in Supplementary Table 2.

### Protein production
The Sema5A ectodomain constructs were produced in HEK 293 T (ATCC CRL-3216) cells at 37 °C. The conditioned medium was collected 5 days post transfection and was buffer exchanged using a QuixStand diafiltration system (GE Healthcare). Proteins were purified by Ni-NTA affinity chromatography (IMAC) (HisTrap FF column, GE Healthcare) followed by size-exclusion chromatography (Superdex 200 increase 10/300 GL or HiLoad 16/60 Superdex 200 pg columns, Cytivia). For crystallization experiments, Sema5A$_{TSR3-4}$-3C-Avi-His$_6$ was produced by transient transfection in HEK293T cells at 37 °C in the presence of the α-mannosidase inhibitor kifunensine to reduce glycan heterogeneity[45]. After the Ni-NTA affinity purification, the protein was incubated with His$_6$-tagged HRV 3C protease at 6.5 °C overnight to remove the His$_6$ tag. The sample was subsequently reloaded to the Ni$^{2+}$-NTA affinity column. The His tag and the HRV 3C protease were trapped by the Ni-NTA affinity column and flow-through fractions of Sema5A$_{TSR3-4}$ were collected to be further purified by size-exclusion chromatography.

### Western blotting for recombinant proteins
Proteins were separated by NuPAGE 4−12% Bis-Tris gels (ThermoFisher Scientific) and transferred to nitrocellulose membranes (Amersham Protran Premium, 0.45 μm). The membranes were blocked with 5% nonfat dry milk (Sema5A$_{TSR3-4}$ proteins with biotinylated Avitag) or 3% Bovine Serum Albumin (His-tagged Sema5A$_{sema-TSR1-7}$ proteins) in PBS for 1 h at room temperature. For His$_6$ tag detection, the membranes were incubated with primary antibody (6xHis Monoclonal Antibody,

TaKaRa, cat. no. 631212 dilution 1:3000) for 1 h at room temperature. Blots were then washed six times for 5 min with PBS-0.1% Tween-20 and incubated for 1 h at room temperature with secondary antibody conjugated to horseradish peroxidase (Anti-mouse IgG peroxidase polyclonal goat antibody, Sigma, cat. no. A0168, dilution 1:10,000). For biotinylated Avitag detection, the blot was incubated with Streptactin HRP (BioRad) antibody at 1:25,000 dilution for 1 h at room temperature. This was followed by washing of blots six times for 5 min with PBS-0.1% Tween-20, and signal detection using ECL (BioRad).

### Heparin affinity chromatography

Purified Sema5A constructs were loaded onto a 5-mL HiTrap heparin HP column (GE Healthcare Life Sciences) equilibrated in 150 mM NaCl, 10 mM HEPES, pH 7.5. Proteins were eluted using a linear NaCl gradient, from 150 mM NaCl, 10 mM HEPES pH 7.5 to 1 M NaCl, 10 mM HEPES pH 7.5 over 10 column volumes with a flow rate of 2 ml/min. Elution was followed by absorption at 280 nm.

### Protein crystallization

Before crystallization, $Sema5A_{TSR3-4}$ was concentrated to 11 mg/ml and treated with 1% endoglycosidase F1 (EndoF1) for 2 h at 37 °C to reduce the N-glycans to single residues. Sitting drop vapour diffusion crystallization trials were set up using a Cartesian Technologies pipetting robot and consisted of 100 nl protein solution and 100 nl reservoir solution[46]. Plates were maintained at 20 °C in a Formulatrix storage and imaging system. We obtained four types of $Sema5A_{TSR3-4}$ crystals with different co-crystallization ligands. The $Sema5A_{TSR3-4}$-$NO_3$ crystals were grown by mixing 11 mg/ml protein solution with a reservoir solution containing 0.1 M BIS-TRIS propane pH 7.0 and 6 M ammonium nitrate; the $Sema5A_{TSR3-4}$-$SO_4$ crystals were grown by mixing 11 mg/ml protein solution with a reservoir solution containing 0.1 M Tris pH 8.5, 0.2 M lithium sulfate and 30% PEG 4000; the $Sema5A_{TSR3-4}$-SOS crystals were obtained from co-crystallization of 11 mg/ml protein solution containing 20 mM sucrose octasulfate (SOS) (Toronto Research Chemicals) with a reservoir solution containing 0.1 M sodium citrate pH 5.0 and 8% PEG 8000; and the $Sema5A_{TSR3-4}$-apo crystals were grown by mixing 8 mg/ml protein solution with a reservoir solution containing 0.1 M sodium citrate pH 5.5, and 38% PEG 200. Crystals were cryoprotected by soaking in reservoir solution supplemented with 25% (v/v) glycerol and then flash-cooled in liquid nitrogen.

### X-ray data collection

Native diffraction data for $Sema5A_{TSR3-4}$ were collected at 100 K at Diamond Light Source I03 and I24 beamlines. Integrated data were obtained for the $Sema5A_{TSR3-4}$-NO3, $Sema5A_{TSR3-4}$-SO4 (high-resolution), $Sema5A_{TSR3-4}$-SOS datasets from AutoPROC[47] implemented in the autoprocessing software pipeline available at Diamond Light Source[48] executing XDS[49], POINTLESS[50], AIMLESS[50], and STARANIZO (https://staraniso.globalphasing.org). Diffraction data obtained for the $Sema5A_{TSR3-4}$-apo crystals were integrated with XDS within the Diamond Light Source autoprocessing software pipeline and merged with XSCALE. To account for the considerable anisotropy of the $Sema5A_{TSR3-4}$-NO3, $Sema5A_{TSR3-4}$-SO4 and $Sema5A_{TSR3-4}$-SOS X-ray datasets, the anisotropic filtering protocol implemented in STARANISO was used to generated datasets that incorporate intensities within a locally averaged value of $I/\sigma(I) > = 1.2$ to define an anisotropic diffraction cut-off surface. Elliptical resolution boundary limits after anisotropy corrections are indicated in Supplementary Table 1.

### S-SAD phasing

As molecular replacement trials of $Sema5A_{TSR3-4}$ datasets using structural homologs have not yielded an appropriate solution, we used single-wavelength anomalous dispersion of S atoms (S- SAD) to solve the phase problem. We collected anomalous diffraction data of $Sema5A_{TSR3-4}$-$SO_4$ crystals at 2.75 Å on the long-wavelength in-vacuum MX beamline I23, Diamond Light Source[51] equipped with a semi-cylindrical Pilatus 12 M detector and multi-axis kappa goniometry. Crystals were harvested on dedicated I23 sample holders, compatible with the in-vacuum cryo-cooling requirements of the I23 sample environment. Samples were transferred to the vacuum environment using an in-house-designed Cryogenic Transfer System. Four datasets of 360 degrees each were collected at a temperature of ~50 K and 2.75 Å wavelength, with a 0.1 s exposure per 0.1-degree rotation, using different kappa goniometer angles. Data were integrated with XDS and merged with XSCALE[49]. The merged reflection file was converted to 'mtz' format with Aimless[50] and further used as input to the phasing pipeline crank2[52] within the CCP4 suite. Substructure detection was done with Prasa[53] using a 3.2 Å resolution cut-off. Automatic model building with Buccaneer[54] yielded a partially complete model (~70%) with 2 molecules in the asymmetric unit.

### Molecular replacement and structure refinement

The S-SAD derived partially complete $Sema5A_{TSR3-4}$-$SO_4$ model was used as a molecular replacement (MR) template in Phaser[55] to solve the isomorphous, higher resolution native $Sema5A_{TSR3-4}$-$SO_4$ structure. This structure was in turn used as a MR template to solve the additional $Sema5A_{TSR3-4}$-$NO_3$, $Sema5A_{TSR3-4}$-SOS and $Sema5A_{TSR3-4}$-apo structures. Right after molecular replacement for the $Sema5A_{TSR3-4}$-SOS structure, a clear electron-dense volume was apparent and it was later identified as the co-crystallization ligand SOS. Geometry optimization and restraint generation for SOS was performed using eLBOW[56] within the PHENIX suite, while restraints for C-mannosyl tryptophan residues were generated within Coot, using ACEDRG[57]. Atomic models were optimized by alternating between refinement with phenix.refine[58], and manual building in Coot[59]. The final refinement of $Sema5A_{TSR3-4}$-$SO_4$ was corroborated by its local superimposition with $Sema5A_{TSR3-4}$-$NO_3$. Data collection and refinement statistics are provided in Supplementary Table 1.

### Structural analysis

Stereochemical properties were assessed in MOLPROBITY[60]. Superpositions were calculated using PDBeFOLD[61] and Pymol (Schrödinger, LLC), electrostatic potentials were generated using APBS[62]. Buried surface areas of protein–protein interactions were calculated using the PDBsum web server[63]. Omit maps of crystallographic ligands were calculated in PHENIX using the Polder map method[64]. Sequence alignments were generated with Clustal Omega[65], and were further edited by Jalview[66]. Structural homologs were identified with the DALI server[67]. Figures were produced with PyMOL (Schrodinger, LLC), ESPRIPT[68] and BioRender (https://biorender.com).

### Rigid body docking

The TSR4 dimer region (residue range: E703-C763 and E703-F771 for the two protein chains) of the $Sema5A_{TSR3-4}$.$NO_3$ structure was used as a protein model for docking after removing hydrogens, solvent molecules, and ligands. The ClusPro[31] web server was used for docking in the fully automated heparin docking mode. A degree of polymerization = 4 (dp4) heparin oligosaccharide was used for docking with the following chemical formula: IdoA(2S)-GlcNS(6S)-IdoA(2S)-GlcNS(6S). Basically, the following computational steps were undertaken by the server. First, rigid body docking was performed by extensive conformational sampling. This was followed by RMSD based clustering of the 1000 lowest energy structures generated to find the largest clusters that will represent the most likely models of the complex. Steric clashes were subsequently removed using energy minimization.

### Alphafold2 structural modeling of $Sema5A_{sema-TSR2}$

The human Semaphorin-5A sema-TSR1-2 domains (Uniprot: Q13591, residue boundaries: E23-P651) was submitted as an input sequence for

ColabFold v1.5.2 notebook[69] for structure prediction of a Sema5A homodimer complex. Five models were generated using the pdb70 template mode and amber molecular dynamics relaxation. The relaxed structure with the highest rank was used to generate a chimera combining Sema5A$_{sema-TSR2}$ in silico and Sema5A$_{TSR3-4}$ crystal structure model.

## GAGOme cell surface binding assays

Sema5A$_{TSR3-4}$ constructs containing a C-terminally fused 3C-Avi-His$_6$ tag were produced in HEK 293T cells, purified from the secreted medium by Ni-NTA, and site-specifically biotinylated at the AviTag conjugated to their C-terminus in vitro, using a BirA biotin-protein ligase standard reaction kit Avidity LLC (CO, USA, avidity.com). Successful biotinylation was confirmed by Western blot incubated with Streptactin HRP (BioRad) antibody at 1:25,000 dilution (Supplementary Fig. 1j). For each assay sample, $1 \times 10^5$ of genetically engineered CHO GS-/- cells[33,34] were harvested and washed in PBS before being resuspended in 50 μg/mL of biotinylated WT, R747E/R749E or K734E/R747E/R749E Sema5A$_{TSR3-4}$ diluted in PBS with added 1% FBS (assay buffer), gently shaking for 1 h at 4 °C. The cells were then washed with assay buffer before incubation with Alexa Flour 488-streptavidin (1:2000, #S32354, Invitrogen) diluted in assay buffer while gently shaking for 30 min at 4 °C. After wash in assay buffer, the cells were resuspended in assay buffer and subjected to flow cytometry on a SA3800 spectral cell analyzer (SONY), where mean fluorescent intensity for each sample was measured. All experiments were performed a minimum of 3 times using triplicate samples, and mean fluorescent intensity was normalized to CHO WT for all samples.

## Biolayer Interferometry

All measurements were performed at 25 °C using streptavidin (SA) biosensors in an Octet Red96e (both Sartorius). We used an assay setup where biotinylated GAGs were immobilized on Streptavidin (SA) coated biolayer interferometry (BLI) biosensor tips and Sema5A$_{sema-TSR1-7}$ was used as the analyte. Unfractionated heparin (Merck) CS-E (Iduron) and CS-A (Merck) were biotinylated at their reducing end using EzLink Biotin LC-hydrazide (Thermo Fisher Scientific, 21340) according to the manufacturer's protocol. The reactions were quenched with 1 M Tris pH 8.0 and extensively dialyzed against PBS to remove un-coupled biotinylation reagent. The analyte buffer included 20 mM HEPES pH 7.5 150 mM NaCl and 0.02% Tween-20. Data were processed and analysed using the Octet Data Analysis HT software (version 11.1). All kinetic data were double referenced. Kinetic traces of Sema5A$_{sema-TSR1-7}$ binding to heparin and CS-E had a complex binding profile that could not be well fitted with a 1:1 binding model. Considering the diverse molecular interactions that together add up to the Sema5A$_{sema-TSR1-7}$ – GAG binding signal, and the heterogeneity of the GAG reagents used as ligands in these assays, the kinetic traces of Sema5A$_{sema-TSR1-7}$- heparin and of Sema5A$_{sema-TSR1-7}$- CS-E interactions were fitted using a 2:1 heterogenous ligand model, that assumes analyte binding at two independent ligand sites. The apparent dissociation constants $K_{D, app}$ -s were determined by fitting the binding response vs analyte concentration plot data with a Langmuir 1:1 binding isotherm in OriginPro 9.1.

## Extraction and purification of HS from CHO Cells

The wildtype CHO cells were washed in PBS and diluted to $1 \times 10^7$ cells/ml in 50 mM TRIS-HCl (pH 7.4), 10 mM CaCl$_2$, and 0.1% Triton X-100. Next pronase (Roche) was added (1 mg/ml), and the reaction was incubated overnight on a rotating tray in an incubator set at 37 °C to digest the cellular proteins. The pronase was heat inactivated at 98 °C for 5 min. MgCl$_2$ (5 mM) and deoxyribonuclease I (1 μg/ml; Sigma-Aldrich) was added, and the sample was incubated at 37 °C for 4 h to digest DNA. The sample was treated with ribonuclease A (10 μg/ml; Sigma-Aldrich) and 5 mM EDTA at 37 °C for 2 h to digest RNA. Next

neuraminidase (0.5 mU/ml; Sigma-Aldrich) was added and incubated at 37 °C overnight to remove sialic acid from the glycans. The sample was then again incubated with pronase at 1 mg/ml for overnight digestion of the proteins at 37 °C. The sample was acidified to pH 4 to 5 with acetic acid, centrifuged at 20,000 g for 20 min and filtered through 0.45 μm filters. The filtered sample was placed onto a HiTrap DEAE FF column (5 ml; GE Healthcare) to purify the HS chains. The DEAE column was equilibrated with 20 mM NaOAc and 0.5 M NaCl (pH 5.0) for five column volumes, the sample loaded onto the column and washed with this buffer with five column volumes and finally the HS chains were eluted with 1.25 M NaCl. HS was precipitated by addition of cold NaOAc-saturated 100% ethanol (3:1, v/v) and centrifuged at 20,000 g for 20 min at 4 °C, and the pellet was dried on a SpeedVac. The HS was resuspended in deionized water, further purified using Discovery BIO Wide Pore C5-5 (Sigma-Aldrich) using deionized water, as the HS chains will not bind to the C5 column and desalted with deionized water using 1-ml HiTrap desalting columns (GE Healthcare).

## Mass photometry

A mass photometry (OneMP, Refeyn Ltd, Oxford, UK) instrument was started and used to measure molecules in the samples. Glass slides (GLW storing systems ZK25) and gaskets (Re-useable Culturewell gasket 3 mm diameter × 1 mm depth, Sigma GBL103250-10EA) were cleaned with water and isopropanol eight times, before being dried with compressed air. A drop of emersion oil (Thorlabs MOIL-30) was placed on the mass photometry objective and a glass slide with a gasket adhered to the glass slide was placed on top of the emersion oil and positioned on the instruments mobile stage. Next, 10 μl of 100 mM NaCl was placed into the gasket, and the mass photometry objective adjusted and focused. The mass photometry instrument was then calibrated by the addition of 10 μl of protein standards of molecular weights 66 kDa, 146 kDa, 242 kDa and 480 kDa from Native Mark Unstained Protein Standard (ThermoFisher LC0725) in 100 mM NaCl. Immediately after the addition of the sample the mass photometry recording was started in Acquire MP. Analysis of each movie was performed with the mass photometry software, DiscoverMP. The contrast measurement of each calibration standard was identified, and presented as histograms with fitted gaussian curves, after this the contrast was converted to masses with this set of standard protein markers. Sema5A ectodomain constructs either with, or without a C-terminal hexahistidine epitope peptide was used to bind to different GAG types. An average molecular mass of 15,000 Da for GAG polymers was used for calculating molar concentrations. Sema5A, 100 nM, was incubated with varying concentrations of heparin (Merck) from 5 μM to 25 nM, size-defined heparin oligosaccharides (Iduron), 400 nM, CS-E (Iduron) from 5 μM to 25 nM and HS extracted from the CHO cells at 400 nM. After calibration the Sema5A samples were performed in the same way. The instrument objective was adjusted and focused on 10 μl of NaCl, then 10 μl of Sema5A incubations were subsequently added to the mass photometry instrument and recordings were started. The incubation of sample concentration is stated, and the final concentration on the mass photometry for measurement of molecules was half of each concentration. The mass photometry mass distributions were plotted with an 8.3 kDa bin size and a best-fit Gaussian distribution. No difference was observed with mass photometry between the Sema5A with and without the C-terminal hexahistidine tag and heparin sulfate, CS or HS.

## Size exclusion chromatography coupled with multi angle light scattering (SEC-MALS)

SEC-MALS experiments were performed using a Wyatt Dawn HELEOS-II 8-angle light scattering detector (with a 663.8 nm laser) and a Wyatt Optilab rEX refractive index monitor linked to a Shimadzu HPLC system comprising LC-20AD pump, SIL-20A autosampler and SPD20A UV/Vis detector. The Sema5A$_{sema-TSR1-7}$ was purified as described above

and its 1 mg/ml samples were incubated alone or with a five-fold molar excess of dp10 and dp30 size-defined heparin oligosaccharides (Iduron) at room temperature prior to MALS analysis. Unfractionated heparin (heparin sodium salt, from porcine intestinal mucosa, Merck, H3149) and CS-A (sodium salt, from bovine trachea, Merck, 27042) were added to Sema5A$_{sema-TSR1-7}$ in equal mg/ml concentration as heparin dp10. SEC-MALS of Sema5A$_{sema-TSR1-7}$ 1 mg/ml, 100 μl alone or mixed with these GAGs was performed using a Superdex 200 10/30 column equilibrated in PBS. Scattering data were analysed and molecular weight was calculated using ASTRA 6 software (Wyatt). Glycosylation of the Sema5A$_{sema-TSR1-7}$ samples was considered during calculation of the dn/dc value. To calculate the total molecular mass of glycoprotein complexes, it was assumed that each of the 11 predicted Asn-linked glycosylation sites of Sema5A$_{sema-TSR1-7}$ was attached to a Man$_9$GlcNAc$_2$ moiety with a mass of 1883 Da. A dn/dc = 0.1770 ml/g value was calculated for Sema5A$_{sema-TSR1-7}$ using dn/dc values for proteins (0.1850 ml/g) and glycans (0.146 ml/g) from[70].

## Isothermal titration calorimetry (ITC)

The interaction of Sema5A$_{sema-TSR1-7}$ to heparin oligosaccharides of defined length, prepared from high grade porcine *heparin* (cf. Materials) was tested by ITC. Experiments were performed using a MicroCal PEAQ-ITC (Malvern) at 25 °C in PBS, with a differential power of 10 μcal s$^{-1}$ and stirring at 750 rpm. Experiments consisted of an initial test injection of 0.4 μL, followed 150 s later by 18 injections of 2 μL, spaced 150 s apart. Sema5A$_{sema-TSR1-7}$ was purified by SEC in PBS and heparin oligosaccharides were dissolved in the SEC buffer. Protein concentrations were determined from the absorbance at 280 nm using calculated molar extinction coefficients, whereas heparin concentration was derived from lyophilized mass. Cell concentrations of 10 μM Sema5A$_{sema-TSR1-7}$ (as disulfide-linked dimer) protein and syringe concentrations of 0.4−2 mM heparin oligos were used in the experiments. Thermograms were integrated and corrected for heats of dilution using PEAQ-ITC analysis software (Malvern). The obtained binding data missed a well-defined initial plateau region that is a characteristic part of the sigmoidal ITC isotherm shape from high-affinity interactions. The shape of a binding isotherm is characterized by the Wiseman "c" parameter[71], and the combination of moderate binding affinity characterized by the dissociation constant ($K_d$), and low concentration (M) of macromolecules used in our experiments, our titrations fall into the low c value regime (0.2−3) according to Eq. 1:

$$c = N^*[M]/K_d \tag{1}$$

Data reliability is still valid at these settings, and, as suggested before[72,73], we fixed the stoichiometry parameter to avoid overparametrization during fitting the data to a one-site equilibrium binding model. Considering that the Sema5A$_{sema-TSR1-7}$ has two GAG sites (N) per disulfide-linked dimer, we attempted data fitting with $N = 2$ and $N = 1$ stoichiometry models and found that for oligosaccharides longer than dp8, only the $N = 1$ model provided a sensible fit of ITC data (cf. Supplementary Fig. 11h−k). Figures were prepared using PEAQ-ITC analysis software (Malvern) and Origin.

## Mice

All procedures involving mice were approved by the University of Michigan Institutional Animal Care and Use Committee under protocol PRO00009791 and performed in accordance with guidelines developed by the National Institutes of Health. All mice were housed under a 12 h light/dark cycle and were given food and water *ad libitum*. *Plxna2$^{-/-}$* mice have been described previously and were maintained on a C57BL/6 background[22,74]. The *Thy1-EGFPm* reporter line (JAX stock #007788) was purchased from the Jackson Laboratory. *Sema5a(R747E_R749E)* mouse line (*Sema5a$^{GAG}$*) was generated by CRISPR/Cas9 mediated gene editing. Briefly, a single strand DNA (sgRNA, 5′-gatccaaatttgctggaagt

agg-3′); and ssODN template (5′- tgagcagcgtttccgctatacctgtaaagct cgcctgccagatccaaatttgctcgaggtcggaGAAcagGAGatagaaatgcggtactgttc cagcgatggaaccagtggctgctccacaga-3′) containing the desired point mutations (AGAcagAGG→GAAcagGAG) to replace Arg747 and Arg749 with glutamic acid residues, were synthesized and purified by Integrated DNA Technologies. In vitro transcription (IVT) for sgRNA was performed with MEGAshortscript T7 kit (Life Technologies) using sgRNA template cloned in px330 plasmid (Addgene plasmid 42230); the IVT product was purified using MEGAclear kit (Life Technologies). A mixture of Cas9 mRNA (TriLink Biotechnologies, 100 ng/μl), sgRNA (50 ng/μl), and ssDNA (100 ng/μl) was injected into fertilized eggs from C57BL/6J mice (JAX stock #000664). Viable two-cell stage embryos were transferred to pseudo-pregnant ICR females to generate six founder mice which were subsequently bred with C57BL/6J mice for germline transmission to generate F1 mice. Five correctly targeted F1 mice were identified and two of these founders were used for further analysis. Mice were maintained on a C57BL/6 background. The presence of the R747E_R749E point mutations in F1 mice was confirmed by DNA sequencing. For PCR genotyping the following primers were used: Forward, 5′- ctgaggctcccctgacatagtgag -3′; Reverse-WT, 5′- ggaac agtaccgcatttctatcctctg -3′; Reverse-Mutant, 5′- ggaacagtaccgcatttct atcTCctgtTC -3′ and amplification conditions: 94 °C for 20 s, 68 °C (repeat for 5 times)/ 67 °C (repeat for 5 times)/ 66 °C (repeat for 23 times)/ for 15 s, 72 °C for 15 s. The size of PCR product for both alleles is 293 bp. Sema5a (R747E_R749E) mice have been donated to Jackson Laboratory and are available upon request.

## Histological procedures

Animal perfusion, tissue preparation, brain sectioning, and staining were carried out as described in ref. 38. Mouse pups at postnatal day (P)14 were subjected to BrdU labeling by *i.p.* injection of BrdU (20 mM, 50 mg/kg. Sigma B9285), 2 h prior to sacrifice. Proliferating cells that incorporated BrdU were detected by immunofluorescence labeling (Rat anti-BrdU, 1:500, Abcam ab6326). For visualization of GFP in *Thy1-EGFPm* brain sections anti-GFP (chicken anti-GFP, 1:500, Aves Lab #GFP-1020) was used. Myelin was stained by FluoroMyelin (1:300, Life technologies F34651) to visualize fasciculus retroflexus (FR). For image acquisition, a Zeiss Axio Observer Z1 equipped with a Zeiss Axiocam 503 mono camera was used. Zen software was used for image tiling and stitching. BrdU$^+$ cells in the dentate were quantified using ImageJ. For the analysis of GFP-labeled dendritic spines in dentate granule cells, we focused on a dendritic segment 50 μm away from the dentate granule cell layer. Spine imaging and quantification was described previously in ref. 22. Images were composed of z-stacks at 0.3 μm intervals, using the Leica S5 confocal microscope with 100x Oil objective plus an extra 1.6x zoom.

## Sema5A Western blot

Forebrains from P7 *Sema5a$^{+/+}$*, *Sema5a$^{GAG/+}$*, and *Sema5a$^{GAG/GAG}$* pups were collected and homogenized in RIPA buffer (0.1% SDS, 1 mM DTT, 135 mM NaCl, 10 mM Tris-HCl pH7.4) supplemented with protease inhibitor cocktail 1:100 (Sigma cat. # P8340). For protein extraction, samples were placed on ice for 30 min and then centrifuged for 20 min at 25,000 g. The supernatant was transferred to a test tube, protein concentration measured, and diluted to 1 mg/ml in lysis buffer. Five ml of lysate were incubated with 50 μl of washed wheat germ agglutinin (WGA)-agarose beads (Vector Laboratories, Cat # AL-1023) and tumbled overnight at 4 °C. Beads were pelleted in an Eppendorf by centrifuging at 800 g for 5 min. The pellet was resuspended in 2 ml of lysis buffer, vortexed, and spun at 800 g for 5 min. This step was repeated five times and the final pellet resuspended in 100 μl 2X Laemmli buffer (Bio-Rad, Cat# 1610737). Western blotting was carried out as described[38] and membranes probed with anti-Sema5A rabbit serum diluted 1:500. To generate anti-Sema5A antiserum, rabbits were immunized with recombinant Sema5A. Briefly, the recombinant rat

Sema5A cytoplasmic region was expressed in *E.coli* and purified by affinity chromatography as described[75]. Antigen was conjugated to KLH and used for immunization of rabbits (Invitrogen Zymed Laboratories). Test bleeds were assayed for reaction with recombinant Sema5A expressed in HEK293T cells. IgGs from reactive sera were purified using a streptavidin column.

### Statistical analysis for mouse genetic data
GraphPad Prism 8 was used for data analysis. A two-tailed unpaired Student's *t* test was used for single comparison, and one-way ANOVA was used for multiple comparisons. $p < 0.05$ was considered statistically significant.

### Reporting summary
Further information on research design is available in the Nature Portfolio Reporting Summary linked to this article.

## Data availability
Coordinates and structure factors have been deposited in the Protein Data Bank with accession numbers: 8CKG (Sema5A$_{TSR3-4}$ complexed with sulfate), 8CKK (Sema5A$_{TSR3-4}$ complexed with nitrate), 8CKL (Sema5A$_{TSR3-4}$ complexed with sucrose octasulfate), and 8CKM (Sema5A$_{TSR3-4}$, unliganded state). All other data needed to evaluate the conclusions in the paper are present in the paper and/or the Supplementary Information. Solution structure of heparin dp18 is available in the Protein Data Bank with accession number 3IRI. The GEO: GSE186216 snRNAseq dataset re-analyzed in this study is available at https://0-www-ncbi-nlm-nih-gov.brum.beds.ac.uk/geo/query/acc.cgi?acc=GSE186216. Source data are provided with this paper.

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

## Acknowledgements

The authors would like to thank Diamond Light Source for beamtime (proposals mx19946 and mx28534), and the staff of beamlines I03, I23 and I24 for assistance with crystal testing and data collection; Weixian Lu for help with tissue culture, Thomas Walter for crystallization technical support, David Staunton for assistance with SEC-MALS experiments. We thank Craig Johnson for analysis of transcriptomics data. We acknowledge support from the Medical Research Council UK (MR/T000503/1 to E.Y.J.), the Carlsberg Foundation (CF20-0412 to R.L.M.), the Novo Nordisk Foundation (NNF22OC0073736 to R.L.M.), the Danish National Research Foundation (DNRF107 to H.C.), the Dr Miriam and Sheldon G. Adelson Medical Foundation (to R.J.G.), the National Institute of Health, (R01 MH119346 to R.J.G.), the Wellcome Trust Award (223133/Z/21/Z to H.C., E.Y.J. and R.L.M.). The Wellcome Centre for Human Genetics is supported by the Wellcome Trust Centre grant 203141/Z/16/Z.

## Author contributions

G.N.N., H.C., R.L.M., R.J.G. and E.Y.J. conceived of and designed the studies and wrote the paper with input from all authors. H.C., R.L.M., R.J.G. and E.Y.J. supervised the project. G.N.N. purified proteins,

crystallized and solved the structures, performed biophysical assays and in silico docking. K.H. helped with protein crystal cryoprotection and native X-ray data collection. R.D., K.E.O. and A.W. performed and/or supervised S-SAD phasing. R.K., R.L.M. and H.C. designed performed, analyzed and/or supervised cell surface GAG binding and mass photometry experiments. X-F.Z., K.W., R.J.G. performed, analyzed and/or supervised mouse developmental genetics experiments.

## Competing interests

The authors declare no competing interests.
