## [Peer Review File · Nature Communications]

Structure and function of Semaphorin-5A glycosaminoglycan interactionsReviewer #1 (Remarks to the Author):

The manuscript by Nagy et al entitled "Structure and function of the Semaphorin-5A glycosaminoglycan interactions" presents a detailed study on the role of glycosaminoglycans (GAG)s in semaphorin signaling. Although it has been known for decades that GAGs can act as modulators of semaphorin signaling in neuronal migration, the molecular mechanisms underlying this process are largely unknown. Nagy et al. present an impressive array of data supporting the idea that certain types of GAGs bind preferentially to Sema5A. In this study, the exclusive GAG binding unit is biochemically identified and structurally characterized. The structural studies reveal that a unique modification in the thrombospondin fold allows for a rigid dimeric structure that is solidified by disulfide bonds. This creates a rigid scaffold for GAG binding, and the binding site is identified using mutagenesis.

The next stage of the study is more convoluted, as the authors try to identify the molecular aspects of GAGs that are required to contribute to semaphoring signaling. There are two aspects to consider: chemical composition and chain length. The authors use an engineered CHO cell line and several knockouts that should lead the cell line to only produce chondroitin sulfate GAGs or heparan sulfate GAGs. All these cell derived GAG preparations involve mixtures, which can be problematic because it only requires a minor fraction of one activating GAG to trigger oligomerization or enhanced binding.

The authors do use purified and enzymatically truncated heparan sulfates to test oligomerization of the Sema5A ectodomain, but they fail to see any oligomerization for this purer GAG material. They only see oligomerization for the heparin mixture (from porcine tissue) and a prep from the CHO cell line, which both still may contain alternative chemical GAG compositions. The authors should therefore tone down their statements that they have shown conclusively that heparan sulfates mediate oligomerization of Sema5A (as stated in the abstract and elsewhere). Unless they can conclusively show that this oligomerization occurs with purified heparan sulfate. Alternatively, they could test oligomerization in the presence of proteoglycans such as glypicans (HS only) or neurexins (CS only) and see if these can trigger oligomerization.

To conclude, I think this manuscript is well suited for Nature Communications as it presents an impressive advancement in the molecular understanding of the role of GAGs in Semaphorin signaling, if the authors can either address the HS oligomerization issue or tone down their statements that they have conclusively shown that HS oligomerization is the only way for GAGs to oligomerize Sema5A.

A few other minor issues:

The data collection statistics for the S-SAD data set is incomplete. It misses "total reflections" as well as most data quality indicators (Rmerge, CC1/2 etc). Please complete the table.

The DALI search results presented in Extended Table 2 are somewhat surprising, claiming there is no homology for TSP4 and distant homology to functionally unrelated structures for TSP3. This could just depend on choosing the domain boundaries. Please send PDB coordinates so this can be verified.

Please also send coordinates for the other structures, so their poor stats and the anisotropic nature of the reflections can be verified.

At the beginning of the Methods section in the main manuscript, cell line culture methods refers to first citation; "...described in [1]" is not appropriate.

Reviewer #2 (Remarks to the Author):

In this manuscript, the authors uncover structural determinants of Sema5a interactions with glycosaminoglycans (GAGs) like heparin sulfate, and identify specific residues which when mutated selectively abolish these GAG interactions. Glycosaminoglycans play important roles in modulating guidance signaling pathways. However, additional work is needed to understand their molecular interactions with and to dissect out their specific contributions to guidance signaling pathways in

vivo. Here, the authors systematically dissect GAG requirements for facilitating Sema5a multimerization. Further, they show in vivo how the Sema5a-GAG interactions play a role in dentate progenitor cell migration.

Overall, the manuscript is well-written and easy to follow. The authors have conducted a comprehensive set of experiments to support most of their claims. Data presented make an important contribution to understanding the effects of GAGs on Sema5a signaling.

Some concerns and suggestions, mostly relatively minor, are listed below:

1-The inability to detect CSPG interactions with TSR3-4 is a limitation of the study. Further discussion or explanation as to why this might be the case is warranted. Might there be a second binding site for CSPG-specific interactions? The GAG mutant was shown to completely disrupt binding to heparin in vitro however was binding to CSPG tested? Might the sema domain be playing a regulatory role? For the CHO cell assay, only TSR3-4 was used, not Sema-TSR1-7, which was used in the previous experiment to demonstrate CSPG interactions. Might that explain the lack of CSPG interaction in the CHO assay? These points should at the very least be discussed, and if there are additional data in hand to address them it should be provided.

2-It would be nice to see some validation that the Sema5a-GAG mutant cannot bind GAGs in vivo. In Supp Fig 11b, it seems that with a WGA pulldown, Sema5a-GAG mutant shows no decreased binding. It is not clear whether or not this figure was supposed to represent a failure to interact with GAGs.

3-Regarding the transheterozygote phenotype of sema5a-GAG and plexA2 mutants, do transhets of Sema5a and plexinA2 also show this same transhet phenotype, or is this specific to the Sema5a-GAG mutant? Rather than a GOF effect, could it reflect some sort of competitive inhibition where reduction of both the ligand and receptor negates the effect? Further, some explanation as to why Sema5a-GAG homozygous mutants were not evaluated together with plexinA2 hets would be welcome. This phenotype could be further informative and if it is not lethal, and would make a nice addition to the manuscript—if data are in hand they should be included, and if not, some mention of why provided.

4-The disease/neurologic disorder-aspect of the paper is mentioned in the abstract, however the data supporting this idea is weak. The ASD mutation identified is not one that was tested in the study. To warrant mention of this in the abstract, the ASD mutation could have been tested at least in vitro in biochemical interaction or folding assays. Since it has not, this aspect of the relationship to disease seems be reserved for the Discussion.

Reviewer #3 (Remarks to the Author):

Reviewer #4 (Remarks to the Author):

It is well established that guidance cues have diverse functions depending on the cellular context, extracellular environment, and receptor binding partners they encounter during development to wire the nervous system. The Sema5A guidance cue has previously been demonstrated to associate with different types of extracellular glycosaminoglycans (GAGs) to exert its attractive and inhibitory effects on axon growth and guidance. However, the mechanism of how Sema5A select the specific GAG to interact with and the functional consequence of its interaction in vivo is not clear. Therefore, this study by Nagy et al. uses x-ray crystallography to reveal for the first time a novel fold configuration in the fourth thrombospondin type-1 repeat (TSR4) of Sema5A provided the structural basis for its GAG binding specificity and demonstrated its functional significance in

vivo should be of interest to the scientific community. The high resolution Sema5A-TSR3-4 crystal structures are beautiful, in combination with the site-directed mutagenesis and in vitro CHO cell experiments convincingly demonstrated the specificity of its binding preference to heparan sulfate proteoglycans (HSPGs). In addition, the mass photometry analysis nicely demonstrated that the interactions with HSPGs could oligomerize Sema5A for signaling with its plexin receptor. In order to show the in vivo consequence of Sema5A GAG interaction in brain development, the authors made a new Sema5A mutant mouse line, Sema5AGAG, using CRISPR/Cas9 gene editing to mutate the GAG binding sites. It is interesting that the Sema5AGAG mutant animals do not display the dendritic spine phenotype seen in hippocampal dentate granule cells of the Sema5A^{-/-} knockout as previously shown, but only the altered number of dentate granule cell progenitors. Finally, the authors demonstrated genetic interaction of Sema5A and Plexin-A2 by generating transheterozygous animals, Sema5AGAG^{+/+};Plexin-A2^{+/-}, to demonstrate that loss of Sema5A GAG interactions altered Plexin-A2 signaling levels to rescue the dentate granule progenitor cell number phenotype. While the authors speculated that the rescued Plexin-A2 phenotype seen in the transheterozygous animals might be due to a Sema5A gain-of-function this was not proven, and which GAG (HSPG or CSPG) is mediating this is not clear. Overall, this study provided novel insights to the specificity of Sema5A GAG interactions through the revelation of a swap fold domain, which is unique to the TSR4 architecture.

While the experiments conducted in this study and their results seem solid, they also raise a few questions, especially those regarding the Sema5AGAG mutant animals, that when addressed will significantly strengthen the mechanistic logic of the Sema5A-GAG binding specificity and the relevance to its in vivo consequence, making this a more complete story.

- 1) There is little to no data provided for the Sema5A/Plexin-A2 signaling mechanism to explain the phenotype of the dentate granule cell number. From what is reported by the authors the Sema5A(R747E;R749E) mutations abolished all GAG interactions with Sema5A, but which GAG is really required for generating proper number of dentate granule cell progenitors? The authors could address which GAG is potentially mediating this by performing immunostaining with anti-Syn3C for HSPGs or anti-CS-56 for CSPGs (previously used in the Kantor et al., 2004, Neuron paper) on wild type hippocampal brain sections. It would be even better if antibodies are compatible to perform double immunostaining with anti-Sema5A and anti-Syn3C or anti-CS-56.
- 2) Is the overall fewer number of BrdU⁺ cells in the dentate gyrus displayed by the Sema5AGAG mutant animals due to a proliferation, differentiation, or migration defect? In this study, the BrdU labeling was only just for 2 hours. However, the authors could increase the time between injection of the BrdU and sacrificing the animals and then double label with anti-Prox1 and anti-BrdU immunostaining, followed by quantifications of BrdU⁺ only versus doubled labeled BrdU + Prox1 cells. The results from this experiment would give some clues to whether the decreased number of BrdU⁺ cells in the Sema5AGAG mutant animals was due to a delay in proliferation or differentiation.
- 3) This bifunctional characteristic of Sema5A signaling through binding with different types of GAGs was first discovered in its ability to attract (when associated with HSPGs) or inhibit (when associated with CSPGs) axons in the fasciculus retroflexus (FR). It is odd that the authors did not mention about this at all in their examination of the Sema5AGAG mutant animals. Do the Sema5AGAG and the Sema5A^{-/-} mutant mice displayed the same FR axon defasciculation phenotype as those observed in the EXT1^{-/-} animals? At the very least, the authors could examine the Sema5AGAG brain sections to determine if guidance phenotypes are present in the FR axons.

POINT BY POINT RESPONSE TO REVIEWERS' COMMENTS

All page and line numbers referred to in the response refer to the revised manuscript where changes are shown.

REVIEWER #1

Remarks to the Author

Query 1. The next stage of the study is more convoluted, as the authors try to identify the molecular aspects of GAGs that are required to contribute to semaphorin signaling. There are two aspects to consider: chemical composition and chain length. The authors use an engineered CHO cell line and several knockouts that should lead the cell line to only produce chondroitin sulfate GAGs or heparan sulfate GAGs. All these cell derived GAG preparations involve mixtures, which can be problematic because it only requires a minor fraction of one activating GAG to trigger oligomerization or enhanced binding. The authors do use purified and enzymatically truncated heparan sulfates to test oligomerization of the Sema5A ectodomain, but they fail to see any oligomerization for this purer GAG material. They only see oligomerization for the heparin mixture (from porcine tissue) and a prep from the CHO cell line, which both still may contain alternative chemical GAG compositions. The authors should therefore tone down their statements that they have shown conclusively that heparan sulfates mediate oligomerization of Sema5A (as stated in the abstract and elsewhere). Unless they can conclusively show that this oligomerization occurs with purified heparan sulfate. Alternatively, they could test oligomerization in the presence of proteoglycans such as glypicans (HS only) or neurexins (CS only) and see if these can trigger oligomerization. To conclude, I think this manuscript is well suited for Nature Communications as it presents an impressive advancement in the molecular understanding of the role of GAGs in Semaphorin signaling, if the authors can either address the HS oligomerization issue or tone down their statements that they have conclusively shown that HS oligomerization is the only way for GAGs to oligomerize Sema5A

Response 1. We thank the Reviewer for pointing out the complexities of analysis of GAGs, we have modified the text to enhance clarity and explain any limitations of the study. While all isolated and cell produced GAGs are inherently heterogenous, we believe that the power of the cell produced GAGs is that specific features on GAGs can be eliminated and determined to be important or not. Thus, our studies with cell derived GAGs involved HS without contaminating CS and *vice versa*, and our dissection of HS features was performed without contaminating CS. In our first submission we showed analysis of HS isolated from CHO cells genetically engineered to lack CS, and we have now expanded this to include additional HS variants. Thus, we believe that the results clearly support our conclusions that extensively sulphated HS support binding and multimerization of Sema5A.

We also show that the Sema5A multimerization effect of heparin is dependent on the chain length of this polysaccharide. This feature is not easily regulated in CHO cells, we therefore

used size fractionated heparin to address this issue (Fig. 4j). We have clarified this point in the text.

Action 1. Fig. 4 has been expanded with additional studies of CHO cell derived HS variants:

Fig. 4. Mass distributions of **f**, Sema5A_{sema-TSR1-7} mixed with HS (1:4) derived from CHO cells with HS3ST1 KI and without CS, **g**, Sema5A_{sema-TSR1-7} mixed with HS (1:4) derived from CHO cells with HS3ST5 KI and without CS; **h**, Sema5A_{sema-TSR1-7} mixed with HS (1:4) derived from CHO cells with NDST2 KI and without CS.

To accommodate this data addition, the data describing Sema5A_{sema-TSR1-7} mixed with heparin dp20 are now moved from Fig. 4 to Supplementary Fig. 9l.

The main manuscript text has been modified as shown below to enhance clarity. Text additions are (highlighted in yellow):

Pages 4-5, line 152: To dissect Sema5A GAG binding specificities, we utilized a genetically engineered library of isogenic Chinese Hamster Ovary (CHO) cells, each with different capacities for GAG biosynthesis due to gene knock-out (KO) and knock-in (KI) of GAG biosynthesis genes (Fig. 3, Supplementary Fig. 1k-l). This CHO library serves as a cell-based display of GAG variants, also referred to as the GAGome^{33,34}, in which the effects of loss/gain of genes on binding of proteins provide indications of the GAG genes and biosynthetic pathways required for the observed binding, and this information may be interpreted to indicate the GAG features involved in binding. CHO WT cells produce rather heterogenous HS and CS GAGs with a repertoire of modifications, and we have previously demonstrated that loss/gain of binding as a result of KO/KI of GAG genes can be used to dissect binding specificities of GAG binding proteins^{33,34}. Flow cytometry binding assays with secreted Sema5A deletion mutants revealed that Sema5A_{TSR3-4} bound to WT CHO cells, while the Sema5A_{TSR3-4} R747E/R749E mutant did not bind (Fig. 3c). Sema5A_{TSR3-4} binding to CHO cells was abolished by KO of *B4galt7*, which completely abrogates both HS and CS biosynthesis (HS+CS KO) by disruption synthesis of the common tetrasaccharide linker on which GAG chains are build (Fig. 3c). The Sema5A_{TSR3-4} binding was shown to require HS and not CS as KO of *Extl3* (HS KO) completely abolished binding, while KO of *CSGalNAcT1/CSGalNAcT2/Chsy1* (CS KO) did not. Since CHO WT cells mainly produce the CS-A variant of CS³³, which in contrast to CS-E did not bind Sema5A_{TSR3-4} in our BLI assay, we also tested Sema5A_{TSR3-4} binding to CHO cells with KI of CHST3 to increase CS 6-O-sulfation³³ and KI of CHST15 (cells producing tiny quantities of CS-E³³), however, the respective binding signals were no greater than that for

CHO WT (Supplementary Fig. 11). These data suggest that either Sema5A does not interact with the set of CSPGs displayed at the CHO cell surface, or, that there is a pronounced preference to bind HS over CS epitopes. To further dissect the HS modifications that support Sema5A_{TSR3-4} binding we targeted *N*-deacetylation/*N*-sulfation by KO of *Ndst1/2* (pre-requisite for further HS modifications³⁵), GlcA/IdoA epimerization by KO of *Glce*, 2-*O*-sulfation by KO of *Hs2st1*, and 6-*O*-sulfation by KO of *Hs6st1/2/3*, which all abrogated binding, thus clearly indicating that the observed binding was to HS (Fig. 3b,c). We also tested CHO cells with enhanced *N*- and 6-*O*-sulfation by KI of NDST2 or HS6ST1, respectively, as well as cells with introduced 3-*O*-sulfation (absent in CHO WT cells) by KI of HS3ST1 and HS3ST5 (Supplementary Fig. 11). Since none of these introduced features affected binding, we conclude that Sema5A binding relies on HS with a complex sulfation pattern of *N*-, 2-*O*-, and 6-*O*-sulfate groups as well as IdoA residues.

Page 5, line 194: The observed Sema5A multimerization was not found with CS-E (Sema5A_{sema-TSR1-7} : CS-E stoichiometry range of ~ 1:50 to 1:0.25) (Fig. 4d, Supplementary Fig. 10a-g). To further study the HS features involved in Sema5A multimerization, we used HS isolated from cell lysates of engineered CHO cells (Fig. 4e-h). HS derived from WT CHO cells clearly supported Sema5A multimerization with the appearance of a mass corresponding to HS with two dimer Sema5A molecules (Fig. 4e). Interestingly, HS isolated from CHO cells with introduced 3-*O*-sulfation (HS3ST1 KI or HS3ST5 KI in CHO cells without contaminating CS (*Csgalact1/2/Chsy1* KO)) appeared to support further multimerization with appearance of a mass corresponding to HS with three Sema5A molecules (Fig. 4g,h). Additionally, HS isolated from CHO cells with KI of NDST2 appeared to reduce multimerization (Fig. 4h). We did not analyse HS from the engineered CHO cells with KO of GAG genes as these did not support binding (Fig. 3). Our results with HS isolated from engineered CHO cells clearly confirm that Sema5A binds HS and can oligomerize on HS chains, further studies are needed to dissect the minimum epitopes.

Text additions to the main manuscript (highlighted in yellow):

Page 5, description of MALS data has been rephrased:

Line 213: Sema5A GAG multimerization analysis in Multi-Angle Light Scattering (MALS) provided consistent results with some multimerization effect observed for the Sema5A_{sema-TSR1-7} +dp30 heparin sample (Fig. 4j).

Minor comments

Query 2. The data collection statistics for the S-SAD data set is incomplete. It misses “total reflections” as well as most data quality indicators (Rmerge, CC1/2 etc). Please complete the table.

Response 2. We thank the reviewer for this comment and have completed the Table S1 with these data.

Action 2. Supplementary Table 1 has been updated with complete S-SAD dataset description.

Dataset	Sema5A _{TSR3-4} -SO ₄ S-SAD	Sema5A _{TSR3-4} -SO ₄ (PDB: 8CKG)	Sema5A _{TSR3-4} -NO ₃ (PDB: 8CKK)	Sema5A _{TSR3-4} apo (PDB: 8CKM)	Sema5A _{TSR3-4} -SOS (PDB: 8CKL)
Data collection	anomalous dataset	native dataset	native dataset	native dataset	native dataset
Beamline	DLS-I23	DLS-I24	DLS-I24	DLS-I24	DLS-I03
Ligand	SO ₄	SO ₄	NO ₃	apo	SOS
Software	XDS/ Aimless	AutoPROC/ STARANIZO	AutoPROC/ STARANIZO	XDS	AutoPROC/ STARANIZO
Reflection scaling method		Anisotropy corrected	Anisotropy corrected		Anisotropy corrected
Wavelength (Å)	2.7552	0.9686	0.9686	0.9999	0.97625
Resolution range (Å)	88.70 - 2.3 (2.38 - 2.3)	88.93 - 1.714 (1.855 - 1.714)	42.75 - 1.557 (1.768 - 1.557)	51.4 - 2.72 (2.77 - 2.72)	41.45 - 2.561 (2.841 - 2.561)
Diffraction limit (Å) along principal axes of fitted ellipsoid		1.75 (0.999a*+0.054c*) 1.64, b* 2.21 (- 0.27a*+0.963c*)	1.94, a* 1.53, b* 2.18, c*		2.625 (0.993a* -0.118c*), 2.42, b*, 3.919 (-0.029a*+c*)
Space group	C 1 2 1	C 1 2 1	P 21 21 21	C 1 2 1	C 1 2 1
Cell dimensions a, b, c (Å) α, β, γ (°)	105.31, 29.25, 90.7 90, 100.01, 90	105.1, 29.3, 90.4 90, 100.5, 90	51.9, 68.3, 85.5 90, 90, 90	104.1, 27.7, 92.3 90, 99.1, 90	106.1, 30.8, 84.2 90, 100.0, 90
Total reflections	226527 (17027)	130867 (3864)	310381 (14907)	44611 (3612)	32914 (1661)
Unique reflections	10568 (854)	20985 (1050)	24530 (1226)	7316 (382)	5008 (250)
Multiplicity	21.4 (19.9)	6.2 (3.7)	12.7 (12.2)	6.1 (5.0)	6.6 (6.6)
Completeness, spherical (%)	85.6 (71.8)	70.7 (16.9)	55.4 (8.9)	98.09 (86.87)	55.9 (10.7)
Completeness, ellipsoidal (%)		89.8 (46.1)	93.8 (72.8)		84.3 (36.6)
Mean I/sigma(I)	17.2 (0.8)	9.3 (1.0)	11.2 (1.9)	7.1 (0.3)	8.2 (0.8)
Wilson B-factor	56.16	29.3	22.67	99.19	73.54
R-merge	0.114 (3.673)	0.096 (1.200)	0.123 (1.564)	0.1335 (5.029)	0.143 (2.670)
R-meas	0.117 (3.769)	0.105 (1.400)	0.128 (1.633)	0.1465 (5.63)	0.156 (2.889)
R-pim	0.024 (0.807)	0.041 (0.700)	0.036 (0.465)	0.05928 (2.46)	0.061 (1.091)
CC1/2	0.999 (0.561)	0.995 (0.338)	0.996 (0.624)	0.994 (0.41)	0.994 (0.098)
Refinement					
No. unique reflections in refinement	10478 (821)	20984 (146)	24526 (141)	7192 (635)	4996
R-work / R-free	0.3814 / 0.4374	0.2060 / 0.2312	0.1966 / 0.2292	0.2878 / 0.3311	0.2554 / 0.2850
No. non- hydrogen atoms		1806	2098	1571	1614
Protein residues		209	234	205	197
RMSDs					
bond length (Å)		0.007	0.003	0.003	0.002
bond angles (°)		0.93	0.70	0.55	0.64
Ramachandran favored (%)		96.0	96.09	95.43	98.40
Ramachandran allowed (%)		4.0	3.91	4.57	1.60
Ramachandran outliers (%)		0	0	0	0
Rotamer outliers (%)		0	0	0	0
Clashscore		2.77	2.48	3.02	6.25
Average B-factor		51.31	35.52	146.20	82.3

Query 3. The DALI search results presented in Extended Table 2 are somewhat surprising, claiming there is no homology for TSP4 and distant homology to functionally unrelated structures for TSP3. This could just depend on choosing the domain boundaries. Please send PDB coordinates so this can be verified.

Response 3. We thank the Reviewer for this comment. In general sense we agree with the notion that the choice of domain boundaries can modulate the structural homology search output. For both TSR3 and TSR4 we used a structural region encompassing three beta strands that comprise the structural unit of thrombospondin type-1 repeats (Tan et al (2002) J. Cell. Biol., reference 31).

The Z-score and RMSD values of the Sema5A TSR3 DALI search results represent a notable (and not distant) structural homology to TSRs from several functionally unrelated proteins, including thrombospondin-1 in which the thrombospondin type-1 repeat has been first described.

For TSR4, we consider that the key to the absence of structural homology to other TSRs is the unique composite TSR4-TSR4' dimer architecture of Sema5A. The TSR4-TSR4' architecture is defined by a beta-strand swap of their third respective strands (see Fig. 1e). The thrombospondin repeat fold typically involves three beta strands in an antiparallel arrangement. For TSR4, its first two strands adopt the usual antiparallel arrangement, however the third strand is directed away from this two-strand unit to make interactions with the same two-strand unit from the other protomer of the Sema5A dimer. A superposition of the Sema5A TSR3 and TSR4 domains made by aligning their respective first two antiparallel beta strands allow direct assessment of their remarkable structural differences. We have included this illustration in the manuscript to complement the current structural description.

Action 3. We repeated the DALI structural homology search for TSR4 with more stringent domain boundaries that include the three beta-strands region (E703-Y754) but excluding a previously considered additional loop region stabilized by an intramolecular disulfide (C755-C763). No structural homology was found for this shorted structural query.

We improved the manuscript by adding interpretations of the DALI search results and by providing illustrations we transformed the Extended Data Table 2 to **Supplementary Fig. 4**. Superimpositions of Sema5A TSR3 with the top five structural homolog domains from unrelated proteins are included as **Supplementary Fig 4 a-e**. A superimposition of Sema5A TSR3 and TSR4 on their first two beta-strands, to show their structural divergence is included as **Supplementary Fig 4f**:

Supplementary Figure 4. Structural homologs of the Sema5A TSR3 domain.

Structural homology search for individual Sema5A TSR3 (Sema5A_{TSR3-4}-NO₃ chain residues 653-702) and TSR4 (Sema5A_{TSR3-4}-NO₃ chain A residues 703-754) domains was carried out using the DALI server²⁵ against the PDB50 database. Structure comparison data with the ten closest structural homologs of Sema5A TSR3 are shown. **a,-e**, Pairwise structural superimpositions of Sema5A TSR3 with the top five structural homolog domains from unrelated proteins. Disulfides stabilizing the TSR fold are shown as sticks. **f**, Structural superimposition of Sema5A TSR3 and TSR4 domains. Note that no structural homologs were retrieved for the TSR4 DALI50 query.

The coordinates of Sema5A_{TSR3-4} NO₃ structure are also uploaded alongside the manuscript to the submission portal to assist manual inspection for structural homology.

Query 4. Please also send coordinates for the other structures, so their poor stats and the anisotropic nature of the reflections can be verified.

Response 4. We thank the reviewer for this comment. The choice of the anisotropic truncation was that this improved the map and B factors during refinement compared to data processing that involved isotropic resolution cut-off based on mean $I / \sigma(I)$ in spherical shells.

Action 4. Coordinates and structure factors for all four Sema5A structures are uploaded separately at the submission portal.

Query 5. At the beginning of the Methods section in the main manuscript, cell line culture methods refers to first citation; "...described in [1]" is not appropriate.

Response 5. We have edited the text.

Action 5. Additional text added to the methods section of the Supplementary material.

Page 2, line 28: Genetically engineered CHO cells, B4GalT7 KO, Csgalnat1/2/Chsy1 KO, Extl3 KO, Ndst1/2 KO, Hs2st1 KO, Glce KO, Hs6st1-3 KO, NDST2 KI, HS3ST1 KI Csgalnat1/2/Chsy1 KO, HS3ST5 KI Csgalnat1/2/Chsy1 KO, CHST3 KI, CHST15 KI were maintained in suspension culture using an equal mix of EX-CELL® CD CHO Serum-Free Medium (#14360C, Sigma-Aldrich) and BalanCD CHO Growth A medium (#91128, Fujifilm), supplemented by 2% GlutaMAX™ (#35050061, Gibco). The cell plasmids and gRNA constructs designed for each cell are described in^{1,2}.

REVIEWER #2

Query 6. The inability to detect CSPG interactions with TSR3-4 is a limitation of the study. Further discussion or explanation as to why this might be the case is warranted. Might there be a second binding site for CSPG-specific interactions? The GAG mutant was shown to completely disrupt binding to heparin in vitro however was binding to CSPG tested? Might the sema domain be playing a regulatory role? For the CHO cell assay, only TSR3-4 was used, not Sema-TSR1-7, which was used in the previous experiment to demonstrate CSPG interactions. Might that explain the lack of CSPG interaction in the CHO assay? These points should at the very least be discussed, and if there are additional data in hand to address them it should be provided.

Response 6. We thank the reviewer for this insightful comment, we have carried out further analysis to address this point. To further dissect Sema5A -CSPG interactions, we assessed the binding of Sema5A_{sema-TSR1-7}, R747E, R749E to surface-immobilized chondroitin sulfate E (CS-E) using biolayer interferometry (BLI). Our results confirmed that the tandem R747E, R749E mutations introduced to Sema5A_{sema-TSR1-7} drastically decreased its capacity to interact with CS-E. We conclude that TSR4 includes the dominant HS and CS GAG binding site of Sema5A, and further structural elements possess minimal direct GAG binding capacity. The herein described experiment also justifies that the TSR3-4 construct is suitable for the CHO cell assays. As for the sema domain, the Supplementary Fig. 1b indicates that it does not have heparin binding capacity. Nevertheless, the sema domain is the functional unit of Sema5A that is used for PlxnA2 interactions (see references: 25, 40).

Action 6. BLI interaction analysis of Sema5A_{sema-TSR1-7}, R747E, R749E with immobilized CS-E is added to the manuscript.

Fig. 3. Chemical specificity of GAGs defines their Sema5A interactions. **a**, BLI binding analysis to characterize the Sema5A interaction with heparin, CS-E, and CS-A. A schematic of the experiment and the calculated apparent K_D values are shown.

Supplementary Figure 1h Bi-layer interferometry titration sensorgrams for characterization of Sema5A R747E, R749E interaction with CS-E

Text addition to the main manuscript:

Page 3, line 145: “Introduction of the R747E/R749E mutations into Sema5A_{sema-TSR1-7} largely abolished CS-E binding”

Query 7. It would be nice to see some validation that the Sema5a-GAG mutant cannot bind GAGs in vivo.

Response 7. We agree with the reviewer. To address this point, we used alkaline phosphatase (AP)-tagged fusion proteins of wildtype Sema5A TSR(1-4) and demonstrate binding to neonatal (P0) mouse brain tissue sections. Strong binding was observed to numerous brain structures, including the developing dentate gyrus in the hippocampus (Supplementary Figure 13f) and the inner retina in the developing eye (Supplementary Figure 13f). These are structures known to require Sema5A function for proper development. Importantly, in the same experiment we included AP-TSR(1-4) in which the GAG binding sites have been mutated (K734E, R747E; R749E). No binding of mutant AP-TSR(1-4) was observed to P0 brain tissue

sections. This demonstrated that mutation of the identified GAG binding site abolishes binding to endogenous GAG present *in vivo*.

Action 7. Text addition to the main manuscript.

Page 4, line 134: To assess whether mutating K734E/R747E/R749E abrogates interaction with endogenous GAGs present in neural tissue, we generated alkaline phosphatase tagged fusion proteins comprised of wildtype or mutated TSR(1-4). Recombinant protein was used for binding to neonatal brain tissue sections. Robust binding was observed for wildtype TSR(1-4), including inner retina, neocortex, and hippocampus. In marked contrast, mutated TSR(1-4) failed to support binding (Supplementary Fig. 12f,g). This demonstrates that K734, R747, and R749 are necessary for binding to GAGs present *in vivo*.

Supplementary Figure 12 f,g: AP-Sema5A TSR1-4 binding to mouse brain section, followed by AP staining, are included in the manuscript.

Supplementary Figure 12. Expression of Sema5A in *Sema5a*^{GAG} mice f, g Binding of recombinant alkaline phosphatase tagged Sema5A thrombospondin repeats 1-4 (AP-Sema5A-TSR1-4) harboring wildtype (WT) or GAG binding deficient (K734E,R747E,R749E, cf. Supplementary Fig. 1d) (G151) TSRs to coronal sections of the P0 mouse head. WT TSR1-4 binds strongly and broadly to tissue sections, including the developing hippocampus (arrow in f), and the inner retina (arrow in g). Scale bar, 1000 µm.

Query 8. In Supp Fig 11b, it seems that with a WGA pulldown, Sema5a-GAG mutant shows no decreased binding. It is not clear whether or not this figure was supposed to represent a failure to interact with GAGs.

Response 8. The WGA pull-down from brain lysates was used to enrich for N-acetylglucosamine linked glycoproteins as a method of purification. This additional purification step is needed because the anti-Sema5A antibodies we generated show high background signals when used for Western blotting of brain lysates. The Sema5A purification step with WGA is unrelated to the Sema5A-GAG interaction.

Query 9. Regarding the transheterozygote phenotype of Sema5a-GAG and plexA2 mutants, do transhets` of Sema5a and plexinA2 also show this same transhet phenotype, or is this specific to the Sema5a-GAG mutant? Rather than a GOF effect, could it reflect some sort of

competitive inhibition where reduction of both the ligand and receptor negates the effect? Further, some explanation as to why *Sema5a*-GAG homozygous mutants were not evaluated together with plexinA2 hets would be welcome. This phenotype could be further informative and if it is not lethal, and would make a nice addition to the manuscript—if data are in hand they should be included, and if not, some mention of why provided.

Response 9. We agree with the reviewer that analysis of *Sema5a*^{+/-};*Plxna2*^{+/-} should be included in the study. We set up additional mouse crosses to obtain *Sema5a*^{+/-};*Plxna2*^{+/-} transheterozygous mice. We find that on a sensitized *Plxna2*^{+/-} background at P14, loss of one *Sema5a* allele (*Sema5a*^{+/-}) results in a rescue of the GC migration phenotype observed in *Plxna2*^{+/-}, essentially mimicking the rescue observed in *Sema5a*^{GAG/+} mice. Based on this additional observation, we propose that the *Sema5a*^{GAG} allele mimics the *Sema5a* null allele (see the revised Figure 5). This suggests that loss of the interaction of Sema5A with GAGs on proteoglycans results in Sema5A loss of function.

Additionally, we attempted to obtain *Plxna2*^{+/-};*Sema5a*^{GAG/GAG} mice. We set up multiple breeding cages over the past 6 months and obtained several litters; however, none of the pups had the desired genotype, suggesting they may die *in utero*.

Action 9.

BrdU+ cell distributions of P14 *Plxna2*^{+/-};*Sema5a*^{+/-} mice included in the manuscript (Fig. 5g,i,j)

Fig. 5g Coronal brain sections through the dorsal DG of P14 *Plxna2*^{+/-};*Sema5a* mice, stained with anti-BrdU.

Fig. 5i, Quantification of total number of BrdU⁺ cells in the DG per tissue section. **j,**

Quantification of BrdU⁺ cells in deep hilar region within the DG. Data are presented as mean ± SEM. **, p<0.01 and ****, p<0.0001. ns, not significantly

Text edits:

We re-assessed the interpretation of the *in vivo* mouse phenotype investigations, based on these additional data, and made some adjustments to our model. Therefore, text changes are applied to the Abstract, Results and Discussion.

Text edits to the Abstract:

Page 1, line 36: We demonstrate that HS-GAG binding is preferred over CS-GAG and mediates Sema5A oligomerization. *In vivo*, Sema5A:GAG interactions are necessary for Sema5A function and regulate Plexin-A2 dependent dentate progenitor cell migration.

Text edits to the Results:

Pages 7, line 268:

Disruption of the Sema5A-GAG interaction results in Sema5A loss of function

Similar to *Sema5a*^{-/-} mice, *Plxna2*^{-/-} mice show defects in dentate progenitor cell migration and distribution within the hilus³⁸. BrdU pulse-labelling of P14 *Plxna2*^{+/-} mice revealed haploinsufficiency for progenitor distribution in the SGZ (Fig. 5f). The total number of BrdU⁺ cells in the dentate is comparable to wildtype mice, however there are fewer BrdU labelled cells in the SGZ and significantly more in the deep hilus (Fig. 5f,i,j). In transheterozygous mice, lacking one allele of *Plxna2* and one allele of *Sema5a* (*Plxna2*^{+/-};*Sema5a*^{+/-}), defects observed in *Plxna2*^{+/-} mice are rescued (Fig. 5g). Because loss of the Sema5A-GAG association may result in Sema5A gain-of-function or loss-of-function, depending on types of HS and CS GAGs in the micro-environment, we wondered whether on a sensitised *Plxna2*^{+/-} background, reduction of the Sema5A-GAG association rescues or worsens defects observed in *Plxna2*^{+/-} mice. To address this question, we generated compound heterozygous mice (*Plxna2*^{+/-};*Sema5a*^{GAG/+}) and assessed distribution of progenitor cells following BrdU pulse labelling (Fig. 5h). Quantification of BrdU⁺ cells in *Plxna2*^{+/-};*Sema5a*^{+/-} and *Plxna2*^{+/-};*Sema5a*^{GAG/+} mice, revealed comparable numbers and distribution, indicating that mutation of the Sema5A GAG binding site results in a loss of function allele. (Fig. 5i,j). Our studies also suggest that the TSR-GAG interaction of Sema5A influences PlxnA2 mediated progenitor cell distribution.

Text edits to the Discussion:

Page 8, line 317:

Mutation of the Sema5A GAG binding site results in a loss-of-function allele

GAGs presented on the cell surface and extracellular matrix may regulate the stoichiometry of receptor complexes for guidance cues during axon pathfinding, however few *in vivo* analyses exist to support this view. One example is motor axon defasciculation by *Drosophila* Sema1A through PlexA, a guidance event that critically depends on the presence of secreted perlecan (encoded by *Hspg2*)⁴³. Accordingly, an important question is whether Sema5A-GAG interactions augment or attenuate Sema5A signalling through PlxnA receptors^{22,38}. We find that

depending on context, dendritic spine density versus progenitor cell migration, disruption of the Sema5A-GAG interaction mimics *Sema5a* loss-of-function. Progenitor cell migration defects observed in *Plxna2*^{+/-} mice are sensitive to reducing the dose of *Sema5a*^{GAG} and *Sema5a*. Loss of either allele rescues defects observed in *Plxna2*^{+/-} mice. Migrating dentate progenitor cells express Sema5A and likely interact with HSPGs and CSPGs on radial glial cells, along which progenitor cells migrate. Based on studies with midbrain neurons, the Sema5A-HSPG interaction is permissive, and the Sema5A-CSPG interaction is non-permissive²⁷. Directed progenitor migration requires the coordinated action of membrane extension, adhesion, translocation, and breaking of adhesion. Disruption of the Sema5A-CSPG interaction may result in too much adhesion and cause progenitor cells to get stuck, conversely loss of the Sema5A-HSPG interaction may result in too much repulsion preventing membrane extension and directional migration, both resulting in reduced cell migration.

Query 10 The disease/neurologic disorder-aspect of the paper is mentioned in the abstract, however the data supporting this idea is weak. The ASD mutation identified is not one that was tested in the study. To warrant mention of this in the abstract, the ASD mutation could have been tested at least in vitro in biochemical interaction or folding assays. Since it has not, this aspect of the relationship to disease seems be reserved for the Discussion.

Response 10 We thank the reviewer for this insightful comment and agree with the need for further analysis. Accordingly, we set out to test the functional effect of the mentioned ASD mutation by introducing it into the Sema5A_{sema-TSR1-7} construct. We performed a small-scale protein expression comparison by transfecting HEK293T with either Sema5A_{sema-TSR1-7} WT or Sema5A_{sema-TSR1-7} R676C, and evaluated proteins secreted into expression medium by anti-His Western blot. We could not detect Sema5A_{sema-TSR1-7} R676C expression. This is possibly a result of decreased structural stability imposed by the R676C residue exchange.

Action 10:

We provide performed a small-scale protein expression study of Sema5A_{sema-TSR1-7} R676C with Sema5A_{sema-TSR1-7} WT as a control followed by western blot analysis. This mutant has failed to express to detectable levels (Supplementary Fig. 2g).

Supplementary Figure. 2g Anti-His Western blot analysis of Sema5A_{sema-TSR1-7}-His₆ (WT) and Sema5A_{sema-TSR1-7}-His₆ R676C (R676C) small-scale protein expression, performed as described in ². Media of transiently transfected HEK293T cells 3 days post-transfection were loaded on SDS-PAGE under reducing conditions. M1: Benchmark Ladder (His-tagged proteins), M2: Benchmark Prestained Ladder (non-His-tagged).

Text addition to the main manuscript:

Page 8, line 352: The R676C dissimilar mutation likely compromises folding, and stability of the protein based on a small-scale protein expression test of Sema5A_{sema-TSR1-7} constructs (Supplementary Fig. 2g).

Details of Sema5A R676C gene synthesis and subcloning are included in the Methods:

Supplementary Material

Page 2, line 58: A construct of human Sema5A Sema5A_{sema-TSR1-7}, (residues 23E-944S), harbouring the R676C mutation was gene synthesized by Genescript and was subcloned into pHLsec vector.

Details of Western blotting for recombinant proteins are included in the Supplementary Methods:

Page 3, line 78:

Western blotting for recombinant proteins

Proteins were separated by NuPAGE 4–12% Bis-Tris gels (ThermoFisher Scientific) and transferred to nitrocellulose membranes (Amersham Protran Premium, 0.45 μm). The membranes were blocked with 5% nonfat dry milk (Sema5A_{TSR3-4} proteins with biotinylated Avitag) or 3% Bovine Serum Albumin (His-tagged Sema5A_{sema-TSR1-7} proteins) in PBS for 1 h at room temperature. For His₆ tag detection, the membranes were incubated with primary antibody (6xHis Monoclonal Antibody, TaKaRa, cat. no. 631212 dilution 1:3000) for 1 h at room temperature. Blots were then washed six times for 5 min with PBS-0.1% Tween-20 and incubated for 1 h at room temperature with secondary antibody conjugated to horseradish peroxidase (Anti-mouse IgG peroxidase polyclonal goat antibody, Sigma, cat. no. A0168,

dilution 1:10,000). For biotinylated Avitag detection, the blot was incubated with Streptactin HRP (BioRad) antibody at 1:25,000 dilution for 1 h at room temperature. This was followed by washing of blots six times for 5 min with PBS-0.1% Tween-20, and signal detection using ECL (BioRad).

REVIEWER #3

(Remarks to the Author):

We thank Reviewers 3 and 4 for their review and insightful comments.

REVIEWER #4

(Remarks to the Author):

It is well established that guidance cues have diverse functions depending on the cellular context, extracellular environment, and receptor binding partners they encounter during development to wire the nervous system. The *Sema5A* guidance cue has previously been demonstrated to associate with different types of extracellular glycosaminoglycans (GAGs) to exert its attractive and inhibitory effects on axon growth and guidance. However, the mechanism of how *Sema5A* select the specific GAG to interact with and the functional consequence of its interaction in vivo is not clear. Therefore, this study by Nagy et al. uses x-ray crystallography to reveal for the first time a novel fold configuration in the fourth thrombospondin type-1 repeat (TSR4) of *Sema5A* provided the structural basis for its GAG binding specificity and demonstrated its functional significance in vivo should be of interest to the scientific community. The high resolution *Sema5A*-TSR3-4 crystal structures are beautiful, in combination with the site-directed mutagenesis and in vitro CHO cell experiments convincingly demonstrated the specificity of its binding preference to heparan sulfate proteoglycans (HSPGs). In addition, the mass photometry analysis nicely demonstrated that the interactions with HSPGs could oligomerize *Sema5A* for signaling with its plexin receptor. In order to show the in vivo consequence of *Sema5A* GAG interaction in brain development, the authors made a new *Sema5A* mutant mouse line, *Sema5AGAG*, using CRISPR/Cas9 gene editing to mutate the GAG binding sites. It is interesting that the *Sema5A* GAG mutant animals do not display the dendritic spine phenotype seen in hippocampal dentate granule cells of the *Sema5A*^{-/-} knockout as previously shown, but only the altered number of dentate granule cell progenitors. Finally, the authors demonstrated genetic interaction of *Sema5A* and *Plexin-A2* by generating transheterozygous animals, *Sema5AGAG*^{+/+};*Plexin-A2*^{+/-}, to demonstrate that loss of *Sema5A* GAG interactions altered *Plexin-A2* signaling levels to rescue the dentate granule progenitor cell number phenotype. While the authors speculated that the rescued *Plexin-A2* phenotype seen in the transheterozygous animals might be due to a *Sema5A* gain-

of-function this was not proven, and which GAG (HSPG or CSPG) is mediating this is not clear. Overall, this study provided novel insights to the specificity of Sema5A GAG interactions through the revelation of a swap fold domain, which is unique to the TSR4 architecture.

While the experiments conducted in this study and their results seem solid, they also raise a few questions, especially those regarding the Sema5A GAG mutant animals, that when addressed will significantly strengthen the mechanistic logic of the Sema5A-GAG binding specificity and the relevance to its in vivo consequence, making this a more complete story.

Query 11. There is little to no data provided for the Sema5A/Plexin-A2 signaling mechanism to explain the phenotype of the dentate granule cell number. From what is reported by the authors the Sema5A(R747E;R749E) mutations abolished all GAG interactions with Sema5A, but which GAG is really required for generating proper number of dentate granule cell progenitors? The authors could address which GAG is potentially mediating this by performing immunostaining with anti-Syn3C for HSPGs or anti-CS-56 for CSPGs (previously used in the Kantor et al., 2004, Neuron paper) on wild type hippocampal brain sections. It would be even better if antibodies are compatible to perform double immunostaining with anti-Sema5A and anti-Syn3C or anti-CS-56.

Response 11. To address which HSPGs and CSPGs in the developing hippocampus may interact with Sema5A, we took advantage of a publicly available single nuclei RNA-sequencing dataset of mouse P10 hippocampus. Mining this dataset for expression of *Sema5a*, *Sema5b*, *Plxna1*, *Plxna2*, *Plxna3*, and *Plxna4* confirmed strong expression of *Sema5a* (but not *Sema5b*) in immature and mature GC (**Supplementary Fig. 13**). Moreover, we observe strong expression of the *Sema5a* receptor *Plxna2* (but not *Plxna1* or *Plxna3*) mostly in immature GC, and to a lower extent in CA3 pyramidal neurons. While numerous HSPGs and CSPGs are expressed in the developing hippocampus, we focused our attention on astrocytes forming the radial glial scaffold along which developing GC are known to migrate within the dentate migratory stream. Our analysis reveals that the glypicans (*Gpc5* and *Gpc6*) and syndecans (*Sdc2* and *Sdc4*) are the most abundant HSPGs and brevican (*Bcan*) and neurocan (*Ncan*) the most abundant CSPGs expressed by astrocytes. Together, these findings suggest that these proteoglycans are candidates that participate in Sema5A interactions (**Suppl. Figure 13**).

Action 11. Supplementary Fig. 13: Single-cell RNAseq dataset from neonatal mouse hippocampus included in the manuscript.

Supplementary Figure 13. Gene expression analysis at single nucleus resolution of the developing mouse hippocampus a, Dot plots analysis of snRNAseq dataset of WT P10 with marker genes used for cell type identification. b, Dot plots analysis of genes of interest reveals

cell type specific distribution and expression levels. Gene expression levels are normalized to average gene expression (color coded calibration). For each cell cluster, the percentile of cells expressing a specific gene product is indicated by the dot size. The snRNAseq dataset (GEO: GSE186216, ⁴³) was re-analyzed. Astrocytes (Astro), neural stem cells (NSC), ependymal cells (EP), immature granulate cells (iGC), mature granule cells (mGC), pyramidal neuron (Py), Cornu Ammonis (CA), excitatory neurons, non-pyramidal (exNeuron), interneuron (IN), Cajal-Retzius cells (CR), presubicular neurons (PS), subicular neurons (SN), oligodendrocyte progenitor cell (OPC), microglia (MG), endothelial cells (EC), pericytes (Peri), mesenchymal cells (Mes), unknown cluster (UK).

Text addition to Results

Page 6, line 256: Gene expression analysis revealed that both, *Sema5a* and *Plxna2* are strongly expressed by immature GC (Supplementary Fig. 13). In addition, several HSPGs (*Gpc5*, *Gpc6*, *Sdc2*, *Sdc4*) and CSPGs (*Bcan*, *Ncan*) are expressed by GFAP⁺ cells that form the radial glial scaffold along which immature GC migrate (Supplementary Fig. 13).

Query 12. Is the overall fewer number of BrdU⁺ cells in the dentate gyrus displayed by the *Sema5A* GAG mutant animals due to a proliferation, differentiation, or migration defect? In this study, the BrdU labeling was only just for 2 hours. However, the authors could increase the time between injection of the BrdU and sacrificing the animals and then double label with anti-Prox1 and anti-BrdU immunostaining, followed by quantifications of BrdU⁺ only versus doubled labeled BrdU + Prox1 cells. The results from this experiment would give some clues to whether the decreased number of BrdU⁺ cells in the *Sema5A*GAG mutant animals was due to a delay in proliferation or differentiation.

Response 12. We refer to a previous study where we reported that in *Plxna2*^{-/-} (and to a lesser extent in *Sema5a*^{-/-} mice) migration of dentate progenitor cells from the ventricular wall into the hilus is significantly delayed compared to parallel processed WT pups (Zhao et al., 2018, PMID: 29320740), arguing that reduced proliferation is a reflection of reduced migration of progenitor into the subgranular zone (SGZ) of the dentate, rather than inability of these cells to proliferate or differentiate.

Regarding the proposed double labelling studies for BrdU incorporated cells and Prox1 (marker for postmitotic GC cells), we agree with the reviewer that this would provide additional evidence that the BrdU⁺ cells are GC progenitors. The position of the BrdU⁺ in the SGZ strongly argues that these indeed are GC progenitors. Finally, we would like to point out that *Sema5a*(GAG/GAG) mice are difficult to obtain, and a substantial number of mice would be needed to address this point experimentally.

Action 12. Text edits to Results:

Page 6, line 254: *Sema5a* and its receptor, *Plxna2*, have previously been shown to regulate progenitor cell migration along the dentate migratory stream and distribution in the subgranular zone (SGZ) of the developing dentate gyrus³⁸.

Line 259: To assess progenitor cell distribution and proliferation in the P14 dentate gyrus of *Sema5a*^{GAG/GAG} mice, we used BrdU pulse-labelling.

Page 7, line 280: Quantification of BrdU⁺ cells in *Plxna2*^{+/-};*Sema5a*^{+/-} and *Plxna2*^{+/-};*Sema5a*^{GAG/+} mice, revealed comparable numbers and distribution, indicating that mutation of the Sema5A GAG binding site results in a loss of function allele. (Fig. 5i,j). Our studies also suggest that the TSR-GAG interaction of Sema5A influences PlxnA2 mediated progenitor cell distribution.

Query 13. This bifunctional characteristic of Sema5A signaling through binding with different types of GAGs was first discovered in its ability to attract (when associated with HSPGs) or inhibit (when associated with CSPGs) axons in the fasciculus retroflexus (FR). It is odd that the authors did not mention about this at all in their examination of the Sema5AGAG mutant animals. Do the Sema5AGAG and the Sema5A^{-/-} mutant mice displayed the same FR axon defasciculation phenotype as those observed in the *EXT1*^{-/-} animals? At the very least, the authors could examine the Sema5AGAG brain sections to determine if guidance phenotypes are present in the FR axons.

Response 13. The Reviewer is correct. Sema5A was originally identified as a bifunctional guidance molecule that participates in FR patterning *ex vivo* (Kantor et al., 2004). We revisited Sema5A function in FR fasciculation taking advantage of *Sema5a*^{-/-} and *Sema5a*(GAG/GAG) mice. We find that neither in *Sema5a*^{-/-} nor in *Sema5a*(GAG/GAG) mice FR fasciculation is defective (**Supplementary Fig. 12** in the revised manuscript). This suggests that FR defects observed *ex vivo* with an anti-Sema5A blocking antibody, and *in vivo* in *Ext1*^{-/-} mice, are not solely due to disruption of Sema5A function.

Action 13. FR analysis from WT, *Sema5a*^{-/-}, and *Sema5a*(GAG/GAG) mouse brains included in the manuscript.

Supplementary Fig. 12c-e Sagittal sections of 3-month-old mouse brains stained with fluoromyelin (green) and DAPI (blue). Rostral is toward the left side. The habenula (Hb) in the dorsal thalamus and the fasciculus retroflexus (FR) are labeled. No obvious defects in FR fasciculation, thickness, or projection toward the interpeduncular nucleus in the midbrain were observed. Representative images of **c** wild-type, **d** *Sema5a*^{-/-}, and **e** *Sema5a*^{GAG/GAG} brains are shown (n= 3 per genotype). Scale bar, 500 μm.

Text edits to Results:

Page 6, line 236: Two independent *Sema5a*^{GAG} lines were established and analysed. Similar to *Sema5a*^{-/-} mice, homozygous *Sema5a*^{GAG/GAG} mice are viable into adulthood, fertile, and at the gross anatomic level, show no obvious defects (**Supplementary Fig. 12c-e**).

Reviewer #1 (Remarks to the Author):

The authors have responded to all the comments from this referee, and I congratulate them with the nice manuscript.

Reviewer #2 (Remarks to the Author):

In this revised manuscript the authors have addressed satisfactorily all of my concerns and are to be commended for such a complete response. There are new data, an explanation here and there, and overall a strengthening of this study. My initial review was positive, and with these revisions I am of the view that this study is now appropriate for publication in Nature Communications.

Reviewer #3 (Remarks to the Author):

Reviewer #4 (Remarks to the Author):

This is a revised manuscript from the authors Nagy et al. I am satisfied with all the revisions they have provided to improved this study. I think the results provide novel insights to the structure and function of Sema5A, and should be of interest to the general scientific community.